# DeepNT: Path-Centric Graph Neural Network for Network Tomography

## Abstract

Network tomography is a crucial problem in network monitoring, where the observable path performance metric values are used to infer the unobserved ones, making it essential for tasks such as route selection, fault diagnosis, and traffic control. Most existing methods require complete knowledge of the network topology and path performance metric calculation formula, which is unrealistic in many real-world practices where network topology and path performance metrics are not well observable. More recently, a few deep learning methods went to the opposite extreme, i.e., turning to data-driven solutions for end-to-end prediction without considering network topology and knowledge of PPMs. In this paper, we argue that a good network tomography requires synergizing the knowledge from both data and appropriate inductive bias from (partial) prior knowledge. To see this, we propose Deep Network Tomography (DeepNT), a new framework that learns a path-centric graph neural network for predicting path performance metrics. The path-centric graph neural network learns the path embedding by inferring and aggregating the embeddings of the sequence of nodes that compose this path. Training path-centric graph neural networks requires learning the network topology and neural network parameters, which motivates us to design a learning objective that imposes connectivity and sparsity constraints on topology and path performance triangle inequality over PPMs. Extensive experiments on real-world and synthetic datasets demonstrate the superiority of DeepNT in predicting performance metrics and inferring graph topology compared to state-of-the-art methods.

## 1 Introduction

Network tomography seeks to infer unobserved network characteristics using those that are observed. More specifically, one may observe path performance metrics (PPMs), such as path delay and capacity, by measuring the two endpoints of the path. Hence, network tomography can use the observations of the PPMs of some pairs of endpoints to infer those of the remaining pairs, because many PPMs can be written as aggregations of corresponding measures on the edges which are typically far fewer the paths they can make up. Network tomography plays a crucial role in applications such as route selection (Ikeuchi et al., 2022; Tao et al., 2024), fault diagnosis (Qiao et al., 2020; Ramanan et al., 2015), and traffic control (Lev-Ari et al., 2023; Pan et al., 2020; Zhang et al., 2018). In real-world applications, many network internal characteristics are not fully accessible. For instance, in scenarios where a local area network is connected to the Internet, the internal computers of the local area network remain hidden from the Internet due to predefined security policies (Fig. 1). This highlights the need to use network tomography to help infer path-related network states such as latency and congestion levels (Cao & Sun, 2012). Similar needs appear in many other fields, such as inferring estimated time of arrival and traffic conditions in transportation networks Zhang et al. (2018), and inferring obscured connections in social networks (Xing et al., 2009).

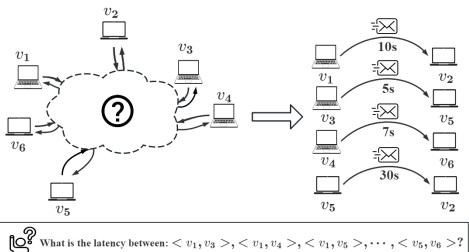

Figure 1: An illustration of network tomography in a sample network, where the end-to-end latency needs to be predicted when the network topology is not available.

Network tomography is very challenging since the PPM values of a pair of nodes are jointly determined by the specific path in a particular network topology under a certain PPM type. To make this problem solvable, traditional network tomography approaches rely on the observed network topology and predefined, hand-crafted PPM calculations, focusing on either additive metrics, where the combined metric over a path is the sum of the involved link metrics (e.g., delay), or non-additive metrics, where the path performance is a nonlinear combination of link metrics (Feng et al., 2020; Xue et al., 2022). Other prescribed methods depend on assumptions like rare simultaneous failures (Carter & Crovella, 1996; Jain & Dovrolis, 2002; Lai & Baker, 2000), minimal sets of network failures (Duffield, 2003; 2006; Kompella et al., 2007), or sparse performance metrics (Firooz & Roy, 2010; Xu et al., 2011; Zhang et al., 2009). These methods rely heavily on human-defined heuristics and rules, making their inference limited and biased by human domain knowledge, especially for many areas where we do not know what PPMs best model the network process. For instance, a heuristic rule that assumes rare simultaneous network failures may be effective for localizing network bottlenecks in a computer network, but would not be suitable for environments like cloud computing or distributed systems, where performance degradation often involves multiple simultaneous disruptions across different nodes or links. More recently, dynamic routing (Sartzetakis & Varvarigos, 2023; Tagyo et al., 2021) and deep learning approaches (Ma et al., 2020; Sartzetakis & Varvarigos, 2022; Tao & Silvestri, 2023) have attempted to bypass the need for prior knowledge on PPMs by directly learning end-to-end models from data (e.g., predicting PPMs given the path's two endpoints). Hence, although they avoided traditional methods' heavy dependency on the observed network topology and prior knowledge of PPMs, they went to the other extreme, by typically completely overlooking the prior knowledge of the PPMs and the intrinsic relation between paths and edges.

To overcome the complementary drawbacks of traditional and deep learning-based methods, we pursue our method, Deep Network Tomography (**DeepNT**), which can infer the network topology and how it determines the PPMs, by deeply characterizing the network process by eliciting and synergizing the knowledge from both training data and partial knowledge of the inductive bias of PPMs. More concretely, we propose a new path-centric graph neural network that can infer the PPMs values of a path by learning its path embedding composed by the inferred sequence of node embeddings along this path. Training path-centric graph neural networks requires learning the network topology and neural network parameters, which motivates us to design a learning objective that imposes connectivity and sparsity constraints on topology and path performance triangle inequality over PPMs. DeepNT addresses two key problems in generic network tomography.

In summary, our primary contributions are as follows:

- **Problem.** We formulate the learning-based network tomography problem as learning representations for end-node pairs to simplify the optimization and identify unique challenges that arise in its real-world applications.

- **Framework.** We propose a novel model for inferring unavailable adjacency matrices and metrics of unmeasured paths, learning end-node pair representations in an end-to-end manner.

- **Adaptivity.** We introduce a novel constrained optimization objective function to infer adjacency matrices by imposing a graph structure constraint and a triangle inequality constraint.

- **Evaluation.** Extensive experiments on real-world and synthetic datasets demonstrate the outstanding performance of DeepNT. DeepNT outperforms other state-of-the-art models in predicting different path performance metrics as well as reconstructing network adjacency matrices.

## 2 RELATED WORK

**Network Tomography** involves inferring internal network characteristics using performance metrics, which can be broadly classified as additive or non-additive. *Additive metrics* frame the network tomography problem as a linear inverse problem, often assuming a known network topology and link-path relationships (Chen et al., 2010; Gurewitz & Sidi, 2001; Liang & Yu, 2003). Statistical methods such as Maximum Likelihood Estimation (MLE) (Teng et al., 2024; Wandong et al., 2011), Expectation Maximization (EM) (Bu et al., 2002; Wandong et al., 2011; Wei et al., 2007), and Bayesian estimation (Wandong et al., 2011; Zhang, 2006) are employed to solve this problem. Algebraic approaches, such as System of Linear Equations (SLE) (Bejerano & Rastogi, 2003; Chen et al., 2003; Gopalan & Ramasubramanian, 2011) and Singular Value Decomposition (SVD) (Chua

et al., 2005; Song et al., 2008), that rely on traceroute work well in certain scenarios but are often blocked by network providers to maintain the confidentiality of their routing strategies. When link performance metrics are sparse, compressive sensing techniques are used to identify all sparse link metrics (Firooz & Roy, 2010; Xu et al., 2011). Furthermore, studies have explored the sufficient and necessary conditions to identify all link performance metrics with minimal measurements (Alon et al., 2014; Gopalan & Ramasubramanian, 2011). *Non-additive metrics*, such as boolean metrics, introduce additional complexity and constraints. These studies often assume that multiple simultaneous failures are rare, focusing on identifying network bottlenecks (Bejerano & Rastogi, 2003; Horton & López-Ortiz, 2003). However, the assumption of rare simultaneous failures is not always valid. Some works address this by identifying the minimum set of network failures or reducing the number of measurements required (Duffield, 2006; Ikeuchi et al., 2022; Zeng et al., 2012). Additionally, several papers have proposed conditions and algorithms to efficiently detect network failures (Bartolini et al., 2020; Galesi & Ranjbar, 2018; He, 2018; Ibraheem et al., 2023), and some studies have attempted to apply deep learning to this field (Ma et al., 2020; Sartzetakis & Varvarigos, 2022; Tao & Silvestri, 2023). However, most existing works rely on hand-crafted rules and specific assumptions, making them specialized for certain applications and unsuitable where prior knowledge of network properties or topology is unavailable.

**GNNs for Graph Structure Learning** can be classified into approaches for learning discrete graph structures (i.e., binary adjacency matrices) and weighted graph structures (i.e., weighted adjacency matrices). Discrete graph structure approaches typically sample discrete structures from learned probabilistic adjacency matrices and subsequently feed these graphs into GNN models. Notable methods in this category include variational inference (Chen et al., 2018), bilevel optimization (Franceschi et al., 2019), and reinforcement learning (Kazemi et al., 2020). However, the non-differentiability of discrete graph structures poses significant challenges, leading to the adoption of weighted graph structures, which encode richer edge information. A common approach involves establishing graph similarity metrics based on the assumption that node embeddings during training will resemble those during inference. Popular similarity metrics include cosine similarity (Nguyen & Bai, 2010), radial basis function (RBF) kernel (Yeung & Chang, 2007), and attention mechanisms (Chorowski et al., 2015). While graph similarity techniques are applied in fully-connected graphs, graph sparsification techniques explicitly enforce sparsity to better reflect the characteristics of real-world graphs (Chen et al., 2020b; Jin et al., 2020). Additionally, graph regularization is employed in GNN models to enhance generalization and robustness (Chen et al., 2020a). In this work, we leverage GNNs to learn end-node pair representations, enabling simultaneous prediction of path performance metrics and inference of the network topology.

## 3 PRELIMINARIES

**Graphs**. A connected network $\mathcal{G}$ is defined as $\mathcal{G} = (V, E)$, where $V$ and $E \subseteq V \times V$ represent the node set and edge set, respectively, let $A$ denote the adjacency matrix of $\mathcal{G}$.

**Path Performance Metrics (PPMs).** Given a graph $\mathcal{G} = (V, E)$, let $\mathcal{P}_{uv} = \{p_{uv}^n\}_{n=1}^N$ represent the set of all possible paths from node $u$ to $v$, where $u, v \in V$, and $N$ denotes the number of possible paths between $u$ and $v$. Let $y_e$ be the performance metric value of an individual edge, where $e \in E$. The path performance metric value is defined as the cumulative performance of all edges on a path, where the cumulative calculation depends on the type of metric being considered. The unified path performance metric is defined as follows:

$$y_{uv}^n = \bigotimes_{e_i \in p_{uv}^n} y_{e_i}, \text{ where } \bigotimes \in \{\sum, \prod, \bigwedge, \bigvee, \min, \max, \cdots\}, \tag{1}$$

where $\bigotimes$ represents an operator that varies based on the type of path performance metrics, such as additive, multiplicative, boolean, min/max, etc. The optimal path performance between two nodes is defined as $y_{uv} = \{y_{uv}^n \mid y_{uv}^n \unrhd y_{uv}^k, \forall p_{uv}^k \neq p_{uv}^n \in \mathcal{P}_{uv}\}$, where $\unrhd$ indicates better performance, depending on the specific type of path performance metric. For instance, in the case of additive metrics such as latency, the operator $\bigotimes = \sum$ and $\unrhd = \leq$, meaning the path performance metric is the sum of latencies along the edges of the path, with lower values indicating better performance. Alternatively, for min metrics like capacity, $\bigotimes = \min$ and $\unrhd = \max$, where the overall path performance is determined by the minimum capacity along the path, and higher values represent better performance.

Figure 2: Overall framework of DeepNT. The network topology is unknown, with only end-to-end additive path performance for some node pairs being observed. We leverage graph neural networks followed by a path aggregation layer to learn the path-centric end-node pair representation. The training of GNNs and the learning of the graph structure are jointly optimized.

**Network Tomography.** Define $T = \{\langle u, v \rangle\}_{u \neq v \in V}$ as the set of all node pairs, where $|T| = \binom{|V|}{2}$. Let $S \subset T$ be a subset of node pairs for which the end-to-end optimal PPMs are measured. The exact path information between any two nodes is unknown. Network tomography aims to use the measured PPMs values in $S$ to predict the end-to-end optimal PPMs value of unmeasured node pairs in $T \setminus S$. The optimal path between two nodes is typically determined by the Best Performance Routing (BPR). For instance, in a computer network, with measured end-to-end transmission delays for certain node pairs $S \subset T$, the goal of network tomography is to infer the minimum delays for the unmeasured pairs in $T \setminus S$, when the exact path information for node pairs in $T \setminus S$ is unknown.

## 4 DEEP NETWORK TOMOGRAPHY

**Overview.** Network tomography is very challenging since the PPM values of a pair of nodes are jointly determined by the specific path in a particular network topology under a certain PPM type. Hence, we propose a DeepNT to jointly infer network topology, consider path candidates, and learn path performance metrics, in order to effectively predict the PPM values of a node pair as shown in Fig. 2. Specifically, we propose a path-centric graph neural network to learn the candidate paths' embedding from the embeddings of the nodes on them and then aggregate them into node pair embedding for the final PPM value prediction, as illustrated in Fig. 2(b) and detailed in Section 4.1. To infer the network topology, DeepNT introduces a learning objective that updates the adjacency matrix of the network by imposing constraints on connectivity and sparsity, as detailed in Section 4.3. This allows for the simultaneous prediction of PPM values and inference of network structure. Moreover, to leverage the inductive bias inherent in different types of PPMs, we introduce path performance triangle inequalities that further refine our predictions, as outlined in Section 4.2.

### 4.1 PATH-CENTRIC GRAPH NEURAL NETWORK

**Candidate paths' information elicitation and encoding.** Since the actual path to measure the PPMs between two nodes is unknown due to partially unknown network topology and process, we aggregate information of multiple promising paths to capture such context. To be specific, for each node pair, e.g., $\langle u, v \rangle$, we leverage BPR to sample $N$ loopless paths between them based on the adjacent matrix $\tilde{A}$, denoted as $\mathcal{P}_{uv}^L = \{p_{uv}^{(n)}\}_{n \in [1,N]}$, ensuring that the lengths do not exceed $L$. Then, the node embeddings of $u$ and $v$ are updated with a path aggregation layer with a permutation-invariant readout function as,

$$e_{vz}^{(n)} = \sigma(r^\top[(h_v, h_z^{(n)})]) \implies \alpha_{vz}^{(n)} = \text{softmax}(e_{vz}^{(n)}), \text{where } z \in p_{uv}^{(n)}, \tag{2}$$

$$\hat{h}_v^{(n)} = h_v + \sigma\left(\sum_{z \in p_{uv}^{(n)}} \alpha_{vz}^{(n)} \cdot h_z^{(n)}\right) \implies \hat{h}_v = \text{READOUT}(\{\hat{h}_v^{(n)}, p_{uv}^{(n)} \in \mathcal{P}_{uv}^L\}), \tag{3}$$

where $\sigma$ indicates the activation function, $r$ is a predefined vector. $\hat{h}_u$ is obtained in the same way. This path-centric embedding aggregates the local neighborhood information of the end-node pair as well as the information in potential optimal paths connecting them. Finally, the concatenated representation of the end-node pair is passed through a projection module to predict the performance metric value $\hat{y}_{uv}$ via $f_\theta : (u, v, \tilde{A}) \rightarrow \hat{y}_{uv}$. The objective function can be formulated as,

$$\min_{\theta, \tilde{A}} \mathcal{L}_{GNN}(\theta, \tilde{A}, Y) = \sum_{u,v \in V} l(f_\theta(u, v, \tilde{A}), y_{uv}), \; s.t., \; \tilde{A} \in \mathcal{A}, \tag{4}$$

where $\theta$ indicates the parameters of $f(\theta)$, $l(\cdot, \cdot)$ is to measure the difference between the prediction $f_\theta(u, v, \tilde{A})$ and the target value $y_{uv}$, e.g., cross entropy for boolean metrics and $l_2$ norm for additive

metrics. Another objective will be introduced in following Section 4.3 to infer a optimal symmetric adjacency matrix $\tilde{A} \in \mathcal{A}$ where $\mathcal{A}$ represents the set of valid adjacency matrices specified in Section 4.3. We then introduce a training penalty that constrains the DeepNT model with a path performance triangle inequality, applicable to any type of path performance metric.

## 4.2 PATH PERFORMANCE TRIANGLE INEQUALITY

Since PPM is for the optimal path among all the paths between the node pairs, the performance of any path between these two nodes cannot exceed the performance of the observed optimal path. For each pair of nodes $u$ and $v$, and for any other node $z$ on the path, the following triangle inequality must hold: $y_{uv} \geqq (y_{uz} \otimes y_{zv})$ because $y_{uv}$, corresponding to the best path between $< u, v >$, should be no worse than $y_{uz} \otimes y_{zv}$, the best path between $< u, v >$ going through $z$. Here, $\geqq$ is a generalized inequality relation that will be specified according to the type of PPM of interest. For example, when the performance metric is *delay* (where better performance corresponds to lower values), $\geqq$ becomes $\leq$. Thus, we will only punish the violation of the above generalized inequality, resulting in the following generalized ReLU style loss given $f_\theta(u, v) = \hat{y}_{u,v}$:

$$\min_{\theta, \tilde{A}} \mathcal{L}_M(\theta, \tilde{A}, Y) = \sum_{u,v \in V} l_M(f_\theta(u, v, \tilde{A}), y_{uz} \bigotimes y_{zv}), \text{ s.t. } \tilde{A} \in \mathcal{A}, \tag{5}$$

where $z$ is a random node in $p_{uv}^{(n)}$, and $p_{uv}^{(n)}$ is the path with the optimal performance in $\mathcal{P}_{uv}^L$. The function $l_M$ computes a penalty enforcing that the predicted performance does not exceed the bounded value, defined as $l_M(\hat{y}, y) = \max(0, \hat{y} \ominus y)$, where $\ominus$ is also chosen adaptively based on the type of path performance metric. As a result, the estimated performance metric is always bounded by the optimal performance among the observed paths.

## 4.3 GRAPH STRUCTURE COMPLETION

Real-world networks, such as social networks, transportation networks, and information networks, are often naturally sparse, noisy, connected, and (partially) unobservable (Fan & Li, 2017; Zhou et al., 2013), which defines the domain $\mathcal{A}$ as specified in the following. To tackle this, we propose to infer the complete graph structure with the graph adjacency matrix $\tilde{A} \in [0, 1]^{|V| \times |V|}$ as follows:

$$\min_{\tilde{A}} \mathcal{L}_S = \|M \odot (\tilde{A} - A)\|_F^2 + \alpha \|\tilde{A}\|_1, \; s.t., \lambda_2(L(\tilde{A})) > \epsilon, \; \tilde{A} \in \mathcal{A}, \tag{6}$$

where $M \in \{0, 1\}^{|V| \times |V|}$ is a matrix with the same size as $A$, a cell of $M$ is equal to 1 if its corresponding edge connectivity (or not) is observed; and 0, otherwise. $\|\cdot\|_F^2$ indicates the Frobenius norm to ensure the new adjacency matrix be close to the observed one, $\|\cdot\|_1$ indicates the $l_1$ norm to remain the new adjacency matrix sparse. $\alpha$ controls the contribution from the sparsity constraint. $\epsilon$ is small positive constant. Network tomography requires the network is connected to ensure that any node within the graph remains reachable from any other node, thus maintaining the graph's utility in representing a communicative or information transfer network (Zhao et al., 2019). Incorporating the connectivity term directly into the objective function introduces non-convexity, complicating optimization by potentially leading to multiple local minima (Ghosh & Boyd, 2006; Kumar et al., 2019). Therefore, we impose it as a constraint to maintain a convex objective function while ensuring global graph connectivity, allowing for more efficient and stable optimization. $\lambda_2(L(\tilde{A}))$ indicates the second smallest eigenvalue of the Laplacian matrix of $\tilde{A}$, which is used to ensure the connectivity (Fiedler, 1973; Zhou et al., 2006).

## 4.4 OPTIMIZATION FOR DEEPNT

We jointly learn the GNN model and the adjacency matrix to infer the optimal network topology for the GNN model on the given task. The final objective function of DeepNT is given as,

$$\arg\min_{\theta, \tilde{A}} \mathcal{L} = \mathcal{L}_{GNN} + \mathcal{L}_S + \gamma \mathcal{L}_M, \; s.t., \lambda_2(L(\tilde{A})) > \epsilon, \; \tilde{A} \in \mathcal{A}, \tag{7}$$

where $\gamma$ is a predefined parameter. Jointly optimizing $\theta$ and $\tilde{A}$ is challenging because it involves navigating a highly non-convex optimization landscape with interdependent variables. The optimization problem in DeepNT is formulated as follows:

$$\mathcal{F} = \min_{\theta, \tilde{A}} g(\theta, \tilde{A}) + \alpha \|\tilde{A}\|_1, \tag{8}$$

where $g(\theta, \tilde{A}) = \mathcal{L}_{GNN} + \gamma \mathcal{L}_M + \|M \odot (\tilde{A} - A)\|_F^2$. Then the proximal gradient algorithm with extrapolation algorithm is shown in Algorithm 1, where $\omega > 0$ is a learning rate, and $\text{prox}_{\lambda\|\cdot\|_1}(f) = S_\lambda(f) = \arg\min_x \left(\frac{1}{2}\|x - f\|_F^2 + \lambda\|x\|_1\right)$ is the soft-thresholding operator.

---

**Algorithm 1:** Optimization of DeepNT

**Require:** $S, y, \omega$.
**Ensure:** Parameters $\theta$ of DeepNT, inferred adjacency matrix $\tilde{A}$.
    Initialize $\tilde{A}^{-1} = \tilde{A}^0 = 0, \theta^{-1} = \theta^0 = 0$.
    **while** Stopping condition is not met **do**
        $\overline{\theta}^k \leftarrow \theta^k + (1 - \omega)(\theta^k - \theta^{k-1})$
        $\overline{A}^k \leftarrow \tilde{A}^k + (1 - \omega)(\tilde{A}^k - \tilde{A}^{k-1})$.
        $\theta^{k+1} \leftarrow \overline{\theta}^k - \omega \nabla g(\overline{\theta}^k, \overline{A}^k)$.
        $\tilde{A}^{k+1} \leftarrow \arg\min_{\tilde{A}}(1/2)\|\tilde{A} - (\overline{A}^k - \omega \nabla g(\overline{\theta}^k, \overline{A}^k))\|_F^2 + \alpha\|\tilde{A}\|_1 = \text{prox}_{\omega\alpha\|\cdot\|_1}(\overline{A}^k - \omega \nabla g(\overline{\theta}^k, \overline{A}^k))$.
        **if** $\lambda_2(L(\tilde{A}^{k+1})) < \epsilon$ **then**
            $\tilde{A}^{k+1} \leftarrow \tilde{A}^{k+1} + \epsilon$
        **end if**
    **end while**
    **return** $\theta$ and $\tilde{A}$.

---

**Theorem 4.1.** *Assume $g(\theta, \tilde{A})$ is Lipschitz continuous with coefficient $l > 0$, and its gradient $\nabla g(\theta, \tilde{A})$ is Lipschitz continuous with coefficient $L > 0$. Let $\frac{1}{L} \leq \omega \leq \sqrt{\frac{L}{L+l}}$, and let $\{(\theta^k, \tilde{A}^k)\}$ be a sequence generated by Algorithm 1, then any of its limit point $(\theta^*, \tilde{A}^*)$ is a stationary point of equation 8.*

This theorem guarantees the convergence of our optimization algorithm. The proof to this theorem is proved in Appendix A.3.

## 5 EXPERIMENT

In this section, we evaluate the effectiveness of DeepNT and compare our approach with state-of-the-art network tomography methods. In addition to the performance in path performance metric prediction, we will also discuss the performance of DeepNT in topology reconstruction.

### 5.1 DATASETS

We conduct experiments on three real-world datasets. These networks include transportation networks, social networks, and computer networks, each with different path performance metrics. Transportation networks are collected from different cities. The social network dataset collects interactions between people on different online social platforms, including Epinions, Facebook and Twitter. The Internet dataset consists of networks with raw IPv6 or IPv4 probe data. The statistics of the real-world datasets are shown in Table 1. Details of data processing and path performance metrics of each dataset are in Appendix A.1.

Table 1: Dataset Statistics: the number of networks, (average) nodes and edges. − indicates the dataset has a singe network.

| Statistics | Internet | | Social Network | | | Transportation | | | | |
|---|---|---|---|---|---|---|---|---|---|---|
| | IPV4 | IPV6 | Epinions | Twitter | Facebook | Anaheim | Winnipeg | Terrassa | Barcelona | Gold Cost |
| Graphs | 10 | 10 | - | - | - | - | - | - | - | - |
| Nodes | 2866.0 | 1895.7 | 75,879 | 81,306 | 4,039 | 416 | 1,057 | 1,609 | 1,020 | 4,807 |
| Edges | 3119.6 | 2221.7 | 508,837 | 1768,149 | 88,234 | 914 | 2,535 | 3,264 | 2,522 | 11,140 |

In addition, we use a synthetic dataset to test the comprehensive performance of our model on networks of different sizes and properties, exploring the robustness and scalability of our model. Synthetic networks are generated using the Erdős-Rényi, Watts-Strogatz and Barabási–Albert models. For network sizes in $50i_{i=1}^{50}$, each graph generation algorithm is used to generate 10 networks with varying edge probabilities for each network size (the edge probability represents the likelihood that any given pair of nodes in the network is directly connected by an edge). We focus on monitor-based sampling scenarios, where some nodes are randomly selected as monitors and the end-to-end path performance between the monitors and other nodes are sampled as training data. We set different sampling rates $\delta \in \{10\%, 20\%, 30\%\}$ to simulate the real network detection scenario (Ma et al., 2020). The sampled path performance is used as training data, that is, $\delta$ of total node pairs are used as training data and the rest of node pairs are used for testing.

Table 2: Mean Absolute Percentage Error (MAPE ↓) and Mean Squared Error (MSE ↓) for **Additive Metrics** on real-world datasets. The best results are highlighted in **bold**. The second best results are underlined. − indicates the model is not able to handle the large network.

| $\frac{|S|}{|T|}$ | Method | Internet | | Social Network | | Transportation | | Synthetic | |
|---|---|---|---|---|---|---|---|---|---|
| | | MAPE ↓ | MSE ↓ | MAPE ↓ | MSE ↓ | MAPE ↓ | MSE ↓ | MAPE ↓ | MSE ↓ |
| 10% | MMP+DAIL | 0.9411 | 143.9463 | - | - | 0.7642 | 41.0764 | 0.6755 | 318.3382 |
| | BoundNT | 0.9250 | 126.2529 | - | - | 0.7183 | 28.4639 | 0.6229 | 244.3805 |
| | Subito | 0.9368 | 129.8293 | - | - | 0.7809 | 39.9722 | 0.6047 | 236.2874 |
| | PAINT | 0.9337 | 130.6045 | - | - | 0.6508 | 27.8604 | 0.3493 | 136.8139 |
| | MPIP | 0.9274 | 125.7296 | - | - | 0.7294 | 28.0741 | 0.6246 | 253.4835 |
| | NeuTomography | 0.8118 | 97.0785 | 1.3872 | 21.5816 | 0.6948 | 30.2343 | 0.3629 | 133.9919 |
| | **DeepNT** | **0.6907** | **84.4514** | **0.8172** | **12.6533** | **0.6342** | **24.0135** | **0.2520** | **79.3843** |
| 20% | MMP+DAIL | 0.8892 | 124.0720 | - | - | 0.6982 | 32.4886 | 0.6324 | 261.3459 |
| | BoundNT | 0.8935 | 120.4519 | - | - | 0.6507 | 27.9272 | 0.5587 | 196.9912 |
| | Subito | 0.9008 | 122.5825 | - | - | 0.6757 | 30.7168 | 0.5571 | 194.0589 |
| | PAINT | 0.8638 | 112.4626 | - | - | 0.5983 | 25.1261 | 0.3091 | 113.6392 |
| | MPIP | 0.8901 | 118.9798 | - | - | 0.6419 | 27.1587 | 0.5663 | 202.6942 |
| | NeuTomography | 0.7547 | 90.1461 | 1.2211 | 18.1076 | 0.6175 | 25.4259 | 0.3315 | 119.6274 |
| | **DeepNT** | **0.6299** | **76.5168** | **0.7593** | **12.0193** | **0.5543** | **21.6331** | **0.2168** | **66.5215** |
| 30% | MMP+DAIL | 0.8219 | 104.2967 | - | - | 0.5839 | 28.1040 | 0.5702 | 218.5386 |
| | BoundNT | 0.8593 | 110.3346 | - | - | 0.5124 | 20.6403 | 0.4772 | 170.3272 |
| | Subito | 0.8466 | 107.0691 | - | - | 0.5493 | 20.6268 | 0.4966 | 169.6914 |
| | PAINT | 0.8108 | 102.0409 | - | - | 0.4629 | 20.0037 | 0.2916 | 92.2561 |
| | MPIP | 0.8532 | 109.7416 | - | - | 0.4905 | 20.1655 | 0.4712 | 171.0216 |
| | NeuTomography | 0.7276 | 84.2087 | 1.1378 | 16.3775 | 0.4433 | 19.1260 | 0.3025 | 97.6045 |
| | **DeepNT** | **0.5842** | **71.0797** | **0.7119** | **10.8074** | **0.3794** | **18.6551** | **0.1935** | **59.0406** |

## 5.2 EVALUATION

**Comparison Methods.** To evaluate the effectiveness of DeepNT, we compare it with the state-of-the-art network tomography methods. Details of the implementation can be found in Appendix A.2.

**MMP+DAIL** (Ma et al., 2013) optimizes additive performance metrics under the assumption of a known network topology and manageable, loop-free routing. **ANMI** (Ma et al., 2015) locates problematic network links by employing a tunable threshold parameter and, given precise metric distributions, further estimates fine-grained link metrics. **AMPR** (Ikeuchi et al., 2022) identifies network states in probabilistic routing environments by adaptively selecting measurements that maximize mutual information. **BoundNT** (Feng et al., 2020) derives upper and lower bounds for unidentifiable links, using natural value bounds to constrain the solution space of the linear system. **Subito** (Tao et al., 2024) formulates a linear system and uses network tomography to estimate link delays with reinforcement learning. **PAINT** (Xue et al., 2022) iteratively estimates and refines link-level performance metrics, minimizing least square errors and discrepancies between estimated and observed shortest paths. **MPIP** (Li et al., 2023) uses graph decomposition techniques and an iterative placement strategy to optimize monitor locations for improved inference of path metrics. **NeuTomography** (Ma et al., 2020) learns the non-linear relationships between node pairs and the unknown underlying topological and routing properties by path augmentation and topology reconstruction.

### 5.2.1 MAIN RESULTS OF PATH PERFORMANCE METRIC PREDICTION

The topological incompleteness is 0.2 (i.e., 20% of the edges are replaced by non-existent edges). For the deep learning models, all experiments are performed 10 times and we report the average accuracy. For the linear system based methods, we adopt the solution from the authors' original implementation. Table 2, Table 3, Table 4 and Table 5 report the results of predicting additive, multiplicative, min/max and boolean path performance metrics, respectively. $\frac{|S|}{|T|}$ means how many node pairs' end-to-end path performance metric values are measured and used for training.

For all types of path performance metrics, DeepNT consistently outperforms all other comparison methods. For additive metrics, although NeuTomography provides the second-best performance in both MAPE (0.8118) and MSE (97.0785) on Internet dataset at the 10% sampling rate, DeepNT significantly outperforms it with a MAPE of 0.6907 and an MSE of 84.4514. The same pattern persists across the 20% and 30% sampling rates. DeepNT exhibits exceptional robustness when faced with different types of network structures (e.g., social, transportation, and synthetic networks), which current models are not well-equipped to handle. For multiplicative metrics, DeepNT's performance remains stable across varying levels of network sparsity, as evidenced by its consistent top rankings in all scenarios. DeepNT achieves a MAPE of 0.0509 and an MSE of 0.0093 on large-scale social networks at sampling rate of 30%, which is much better than NeuTomography's MAPE of 0.0813 and MSE of 0.0226, while other models cannot handle these large-scale networks.

**Scalability Analysis.** As the network size increases, DeepNT shows superior scalability compared to the comparison models. For instance, for additional metrics on small datasets (e.g., the transporta-

Table 3: Mean Absolute Percentage Error (MAPE ↓) and Mean Squared Error (MSE ↓) for **Multiplicative Metrics** on real-world datasets. The best results are highlighted in **bold**. The second best results are underlined. * indicates logarithmic transformations are used to convert multiplicative metrics to additive metrics, as these methods are designed for additive metrics.

| $\frac{|S|}{|T|}$ | Method | Internet | | Social Network | | Synthetic | |
|---|---|---|---|---|---|---|---|
| | | MAPE ↓ | MSE ↓ | MAPE ↓ | MSE ↓ | MAPE ↓ | MSE ↓ |
| 10% | BoundNT (*) | 0.3930 | 14.4673 | - | - | 0.0783 | 0.3651 |
| | MPIP (*) | 0.4007 | 16.4387 | - | - | 0.0791 | 0.3583 |
| | NeuTomography | 0.0632 | 0.4216 | 0.0988 | 0.0357 | 0.0347 | 0.0969 |
| | **DeepNT** | **0.0243** | **0.0438** | **0.0620** | **0.1247** | **0.0182** | **0.0154** |
| 20% | BoundNT (*) | 0.3797 | 13.0014 | - | - | 0.0787 | 0.3301 |
| | MPIP (*) | 0.3835 | 12.5421 | - | - | 0.0803 | 0.3498 |
| | NeuTomography | 0.0587 | 0.3493 | 0.0939 | 0.0341 | 0.0291 | 0.0664 |
| | **DeepNT** | **0.0207** | **0.0257** | **0.0571** | **0.1094** | **0.0136** | **0.0087** |
| 30% | BoundNT (*) | 0.3606 | 10.9892 | - | - | 0.0740 | 0.3125 |
| | MPIP (*) | 0.3588 | 12.8381 | - | - | 0.0753 | 0.3216 |
| | NeuTomography | 0.0526 | 0.2409 | 0.0813 | 0.0226 | 0.0243 | 0.0381 |
| | **DeepNT** | **0.0169** | **0.0093** | **0.0509** | **0.0093** | **0.0112** | **0.0083** |

Table 4: Mean Absolute Percentage Error (MAPE ↓) and Mean Squared Error (MSE ↓) for **Min or Max Metrics** on real-world datasets. The best results are highlighted in **bold**. The second best results are underlined.

| $\frac{|S|}{|T|}$ | Method | Internet | | Transportation | | Synthetic | |
|---|---|---|---|---|---|---|---|
| | | MAPE ↓ | MSE ↓ | MAPE ↓ | MSE $(\times 10^6)$ ↓ | MAPE ↓ | MSE ↓ |
| 10% | ANMI | 0.0907 | 52.2783 | 1.0674 | 83.3370 | 0.0975 | 70.4885 |
| | NeuTomography | 0.0741 | 37.8309 | 1.1983 | 114.1017 | 0.0770 | 51.1739 |
| | **DeepNT** | **0.0640** | **34.4012** | **0.4744** | **38.1392** | **0.0585** | **29.7446** |
| 20% | ANMI | 0.0929 | 50.2317 | 1.1205 | 87.6417 | 0.0973 | 67.7634 |
| | NeuTomography | 0.0596 | 28.8063 | 0.9016 | 73.8099 | 0.0633 | 33.1365 |
| | **DeepNT** | **0.0517** | **22.7196** | **0.5216** | **28.5838** | **0.0431** | **21.7088** |
| 30% | ANMI | 0.0944 | 52.5456 | 1.0233 | 84.9310 | 0.0911 | 70.6701 |
| | NeuTomography | 0.0552 | 21.2428 | 0.8278 | 55.4812 | 0.0468 | 18.2206 |
| | **DeepNT** | **0.0396** | **14.2014** | **0.4863** | **22.5173** | **0.0332** | **12.9561** |

Table 5: Accuracy (ACC in % ↑) and $F_1$ Score ↑ for **Boolean Metrics** on real-world datasets. The best results are highlighted in **bold**. The second best results are underlined. − means that the method cannot handle the network.

| $\frac{|S|}{|T|}$ | Method | Social Network | | Transportation | | Synthetic | |
|---|---|---|---|---|---|---|---|
| | | ACC ↑ | $F_1$ ↑ | ACC ↑ | $F_1$ ↑ | ACC ↑ | $F_1$ ↑ |
| 10% | AMPR | - | - | 0.7059 | 0.6184 | 0.6299 | 0.6178 |
| | NeuTomography | 0.6429 | 0.6838 | 0.7858 | 0.7493 | 0.6784 | 0.7226 |
| | **DeepNT** | **0.6854** | **0.7122** | **0.8144** | **0.8003** | **0.7383** | **0.7709** |
| 20% | AMPR | - | - | 0.7517 | 0.6361 | 0.6497 | 0.6246 |
| | NeuTomography | 0.6826 | 0.7148 | 0.8092 | 0.7652 | 0.6980 | 0.7837 |
| | **DeepNT** | **0.7045** | **0.7273** | **0.8317** | **0.8117** | **0.7726** | **0.8163** |
| 30% | AMPR | - | - | 0.7696 | 0.6605 | 0.6707 | 0.6422 |
| | NeuTomography | 0.7213 | 0.7484 | 0.8551 | 0.7795 | 0.7426 | 0.7932 |
| | **DeepNT** | **0.7539** | **0.7691** | **0.8784** | **0.8450** | **0.8063** | **0.8361** |

tion dataset), comparison methods achieve comparable performance to DeepNT. At 10% sampling, NeuTomography achieves a MAPE of 0.6948, close to DeepNT's 0.6342, while PAINT records a MAPE of 0.6508. This shows that on small networks, traditional models can be comparable to DeepNT. However, as the network size increases, such as on social network and Internet datasets, the performance gap between DeepNT and these comparison methods becomes obvious. On the Internet dataset at 30% sampling rate, DeepNT achieves a MAPE of 0.5842, while NeuTomography lags behind with a MAPE of 0.7276. PAINT only achieves a MAPE of 0.8108 on the same dataset.

### 5.2.2 CASE STUDY OF NETWORK TOPOLOGY RECONSTRUCTION

We further analyze the performance of network topology reconstruction given limited path information. We present a case study demonstrating the effectiveness of our proposed method for reconstructing network topology. For a network with 1,000 nodes and 2,521 edges with a topological error rate of 0.2, we visualize the heatmap of adjacency matrices where each block contains 50 nodes.

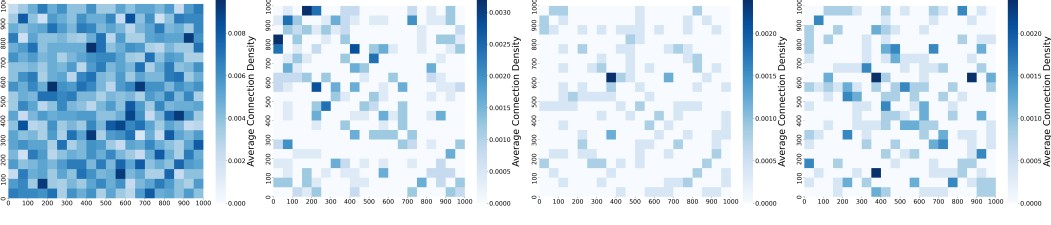

| (a) True topology | (b) Difference between observed topology and real one | (c) Difference between learned topology by DeepNT and real one | (d) Difference between learned topology by NeurTomography and real one |

Figure 3: Heatmap of (a) the true adjacency matrix; heatmap of the difference between the true adjacency matrix and (b) the observed incomplete adjacency matrix, (c) the adjacency matrix learned by DeepNT, and (d) the adjacency matrix learned by NeurTomography, for a synthetic network with 1,000 nodes and 2,521 edges, with a topological error rate of 0.2 and path performance metrics, i.e., bandwidth.

The path performance metric is the min/max metric, i.e., bandwidth, and $|S|/|T| = 30\%$. As shown in Fig. 3, DeepNT successfully recovers most of the true topology with minimal deviation

in topology reconstruction. The heatmaps in Figs 3b, 3c, 3d demonstrate that the adjacency matrix learned by DeepNT is closer to the true adjacency matrix than the observed adjacency matrix and the adjacency matrix learned by NeuTomography. In particular, the denser and more complex parts of the network are more accurately recovered by DeepNT, which leads to smaller differences with the true adjacency matrix.

### 5.3 Ablation Study

To better understand how different components help our model predict various path performance metrics with incomplete network topology, we conduct ablation studies under different topology error rates $\Delta$ when $|S|/|T| = 30\%$. There are two key predefined parameters, i.e., $\alpha$ and $\gamma$, which control the contributions for sparsity and path performance bounds, respectively. We set the value of one parameter to one and the others to zero, and examine how the performance changes to show the impact of each component.

Accordingly, two model variants, DeepNT-$\alpha$ and DeepNT-$\gamma$, are introduced. DeepNT-$\alpha$ sets $\alpha$ to $10^{-4}$ and $\gamma$ to 0, while DeepNT-$\gamma$ sets $\gamma$ to 1 and $\alpha$ to 0. We average the results of 500 Erdős-Rényi networks of the synthetic dataset for various tasks. *Regression* represents the average results for predicting additive, multiplicative, and min/max path performance metrics, while *Classification* reports the average results for predicting boolean path performance metrics. As shown in Table 6, when the sparsity ($\alpha$) or path performance bound ($\gamma$) constraints are removed, the performance significantly drops, demonstrating the importance of sparsity and boundary constraints under incomplete topological information. When the topology error rate $\Delta$ is small, DeepNT-$\gamma$ does not significantly improve the prediction performance. However, when $\Delta$ becomes larger, DeepNT-$\gamma$ (i.e., the path performance bound) can effectively reduce the impact of incorrect topology on prediction because it utilizes the possible correct path information to reconstruct the topology. Additionally, as $\Delta$ increases, the performance gap between DeepNT-$\alpha$ and DeepNT-$\gamma$ narrows, suggesting that maintaining sparsity in the adjacency matrix improves the model's performance lower bound.

Table 6: Ablation study results.

| $\Delta$ | Method | Regression MAPE $\downarrow$ | Classification ACC $\uparrow$ | Classification $F_1$ $\uparrow$ |
|---|---|---|---|---|
| 10% | DeepNT | 0.0741 | 0.8124 | 0.8043 |
| | DeepNT-$\alpha$ | 0.1236 | 0.6909 | 0.7341 |
| | DeepNT-$\gamma$ | 0.1014 | 0.7375 | 0.7582 |
| 20% | DeepNT | 0.0852 | 0.7701 | 0.8041 |
| | DeepNT-$\alpha$ | 0.1305 | 0.6628 | 0.6733 |
| | DeepNT-$\gamma$ | 0.1187 | 0.7081 | 0.7336 |
| 30% | DeepNT | 0.1129 | 0.7226 | 0.7636 |
| | DeepNT-$\alpha$ | 0.1368 | 0.6490 | 0.6827 |
| | DeepNT-$\gamma$ | 0.1212 | 0.6905 | 0.7294 |

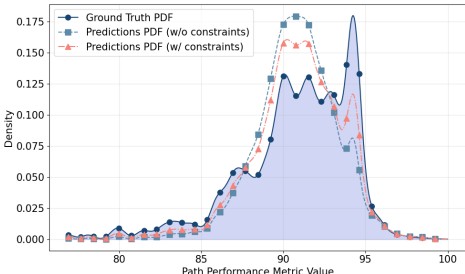

Figure 4: Distribution of the ground truth and predicted min/max path performance metric value by DeepNT, i.e., bandwidth in an IPV4 network under $|S|/|T| = 10\%$.

To further demonstrate the impact of removing constraints on performance, we present the distribution of ground truth and predicted values for the min/max path performance metric (bandwidth) in an IPv4 network with a topological error rate of 30% and $|S|/|T| = 10\%$. As shown in Fig. 4, even under a high topology error rate and limited path information, the probability distribution of predicted path performance metrics by DeepNT closely aligns with the true distribution, highlighting the model's effectiveness. Moreover, the predictions by DeepNT with constraints are closer to the true distribution compared to the variant without constraints, demonstrating the effectiveness of our training framework.

## 6 Conclusion

In this paper, we introduce DeepNT, a novel framework for network tomography that addresses key challenges in predicting path performance metrics and network topology inference under incomplete and noisy observations. Through comprehensive experiments on real-world and synthetic datasets, DeepNT consistently outperforms state-of-the-art methods across a variety of path performance metrics, including additive, multiplicative, min/max, and boolean metrics. DeepNT demonstrates strong scalability and robustness, particularly as the network size and complexity increase, where traditional methods struggle to maintain performance. Additionally, the ablation studies validate the critical role of the proposed constraints on improving prediction accuracy, especially in high-topology error scenarios. Future work will focus on enhancing the adaptability of DeepNT to more complex performance metrics and network environments, including dynamic and multi-layered networks.

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

# A APPENDIX

## A.1 DATASETS

For the initialization representation of nodes, if the source data already contains node features, we use them as the initialization node representations, otherwise we use binary encoding to convert the node identifier (e.g., node index) into a binary representation. For path performance metrics, we use the edge labels of the source data to generate the path labels. In addition, we generate random edge labels of other path performance metric types for some networks, and then generate the path labels. Path performance metrics of each dataset are shown in the Table 7.

Table 7: Properties of datasets. ✓ of binary encoding indicates that the original data has no node features, and we use binary encoding to generate the initial node representation. ✗ means that binary encoding is not used, but the node features of the original data are used. For the path performance metrics, ✓ for one metric indicates that the network has the true edge (link) labels of that metric, while ✓ indicates that random edge labels are generated for that performance metric.

| Properties | Internet[1] | | Social Network[2] | | | Transportation[3] | | | | |
|---|---|---|---|---|---|---|---|---|---|---|
| | IPV4 | IPV6 | Epinions | Twitter | Facebook | Anaheim | Winnipeg | Terrassa | Barcelona | Gold Cost |
| Binary Enc. | ✓ | ✓ | ✗ | ✗ | ✗ | ✓ | ✓ | ✓ | ✓ | ✓ |
| Additive Path Performance Metrics | | | | | | | | | | |
| Delay | | | ✓ | ✓ | ✓ | | | | | |
| RTT | ✓ | ✓ | | | | | | | | |
| Distance | | | ✓ | ✓ | ✓ | | | | | |
| Flow Time | | | | | | ✓ | ✓ | ✓ | ✓ | ✓ |
| Multiplicative Path Performance Metrics | | | | | | | | | | |
| Reliability | ✓ | ✓ | | | | | | | | |
| Trust Decay | | | ✓ | ✓ | ✓ | | | | | |
| Min or Max Path Performance Metrics | | | | | | | | | | |
| Bandwidth | ✓ | ✓ | | | | | | | | |
| Capacity | | | | | | ✓ | ✓ | ✓ | ✓ | ✓ |
| Boolean Path Performance Metrics | | | | | | | | | | |
| Is Trustworthy | | | ✓ | | | | | | | |
| Is Secure | | | | | | ✓ | ✓ | ✓ | ✓ | ✓ |

For the synthetic dataset, we utilize the Erdos-Renyi algorithm to generate networks of different sizes. Then, we generate random edge labels for all the above path performance metrics. When generating random edge labels for all datasets, for the addition and min/max metrics, the random labels are in $[1, 100]$, and for the multiplication metric, the random labels are in $[0.9, 0.999]$. For the Boolean metrics, each edge is randomly assigned a state of 0 or 1, where path labels are controlled to be balanced (the number of positive and negative labels will not less than 30%).

## A.2 IMPLEMENTATION

The training data for each dataset depends on the sampling rate, i.e., $\frac{|S|}{|T|}$ which indicates how many node pairs' end-to-end path performance metric values are used for training. Half of node pairs in $S$ is used as training data, and the other half is used as the validation set. That is, the number of node pairs actually used as training data is $\frac{|S|}{2}$. The rest of node pairs in $T \setminus S$ are used as test data.

To train DeepNT, we use CrossEntropyLoss as the loss function for classification tasks (Boolean metric) and MSELoss as the loss function for regression tasks (additive, multiplicative, and min/max metrics). Adam optimizer is used to optimize the model. The learning rate is set to 1e-4 across all tasks and models. The training batch is set to 1024 and the test batch is 2048 for all datasets. We use GCN as the GNN backbone, and the number of layers of GCN is 2. We use the mean pooling as the READOUT function. An one-layer MLP is used to make predictions. All models are trained for a maximum of 500 epochs using an early stop scheme with the patience of 10. The hidden dimension is set to 256. The hyperparameters we tune include the number of sampled shortest paths $N$ in 1, 2, 3, the sparsity parameter $\alpha$ in 10e-5, 10e-4, 10e-3, 10e-2, and the path performance bound parameter $\gamma$ in 0.1, 0.25, 0.5, 1, 2, 4, 8, 16.

For the comparison methods, we follow the original settings provided by the authors. In particular, the ANMI threshold ratio is reported as 30%. AMPR requires multiple probe tests, and we set the number of probe tests to the number of placed monitors, since each monitor tests the end-to-end path performance with other nodes.

---

[1] https://publicdata.caida.org/datasets/topology/ark
[2] https://snap.stanford.edu/data
[3] https://github.com/bstabler/TransportationNetworks

### A.3 PROOF OF THEOREM 4.1

*Proof.* With the chosen step size satisfying $\frac{1}{L} \leq \omega \leq \sqrt{\frac{L}{L+l}}$, the proximal gradient algorithm ensures:

$$\mathcal{F}(\theta^{k+1}, \tilde{A}^{k+1}) \leq \mathcal{F}(\theta^k, \tilde{A}^k).$$

This implies that the sequence $\{\mathcal{F}(\theta^k, \tilde{A}^k)\}$ is non-increasing. Since $\mathcal{F}(\theta, \tilde{A})$ is bounded below (due to the coercivity of the $\ell_1$-regularization term $\alpha\|\tilde{A}\|_1$), the sequence $\{\mathcal{F}(\theta^k, \tilde{A}^k)\}$ converges to a finite value.

The boundedness of $\mathcal{F}(\theta^k, \tilde{A}^k)$ ensures that the sequence $\{(\theta^k, \tilde{A}^k)\}$ is bounded, which means $\{(\theta^k, \tilde{A}^k)\}$ has at least one limit point $(\theta^*, \tilde{A}^*)$.

Since $\|(\theta^{k+1}, \tilde{A}^{k+1}) - (\theta^k, \tilde{A}^k)\| \to 0$, and $\nabla g$ is Lipschitz continuous, it follows that:

$$\|\nabla g(\theta^{k+1}, \tilde{A}^{k+1}) - \nabla g(\theta^k, \tilde{A}^k)\| \to 0.$$

Therefore, the gradients $\nabla g(\theta^k, \tilde{A}^k)$ converge to $\nabla g(\theta^*, \tilde{A}^*)$ as $k \to \infty$.

The update for $\theta$ in Algorithm 1 is:

$$\theta^{k+1} - \overline{\theta}^k = -\omega\nabla_\theta g(\overline{\theta}^k, \overline{A}^k).$$

As $k \to \infty$, $\overline{\theta}^k \to \theta^*$, $\theta^{k+1} - \overline{\theta}^k \to 0$ and $\nabla_\theta g(\overline{\theta}^k, \overline{A}^k) \to \nabla_\theta g(\theta^*, \tilde{A}^*)$. Then, it follows that:

$$\nabla_\theta g(\theta^*, \tilde{A}^*) = \mathbf{0}.$$

The update for $\tilde{A}$ in Algorithm 1 involves solving the proximal operator:

$$\tilde{A}^{k+1} = \arg\min_{\tilde{A}} \left( \frac{1}{2} \left\| \tilde{A} - \left( \overline{A}^k - \omega\nabla_{\tilde{A}} g(\overline{\theta}^k, \overline{A}^k) \right) \right\|_F^2 + \omega\alpha\|\tilde{A}\|_1 \right).$$

This optimization is equivalent to applying the proximal mapping:

$$\tilde{A}^{k+1} = \text{prox}_{\omega\alpha\|\cdot\|_1} \left( \overline{A}^k - \omega\nabla_{\tilde{A}} g(\overline{\theta}^k, \overline{A}^k) \right),$$

where $\text{prox}_{\lambda\|\cdot\|_1}(f) = S_\lambda(f)$ is the soft-thresholding operator. The proximal mapping satisfies the optimality condition:

$$\mathbf{0} \in \tilde{A}^{k+1} - \left( \overline{A}^k - \omega\nabla_{\tilde{A}} g(\overline{\theta}^k, \overline{A}^k) \right) + \omega\alpha\partial\|\tilde{A}^{k+1}\|_1.$$

Rearranging this condition gives:

$$\mathbf{0} \in \nabla_{\tilde{A}} g(\overline{\theta}^k, \overline{A}^k) + \frac{1}{\omega}(\tilde{A}^{k+1} - \overline{A}^k) + \alpha\partial\|\tilde{A}^{k+1}\|_1.$$

As $k \to \infty$, the extrapolated sequence $\overline{A}^k \to \tilde{A}^*$ and the proximal updates $\tilde{A}^{k+1} \to \tilde{A}^*$. Consequently, the term $(\tilde{A}^{k+1} - \overline{A}^k)/\omega \to \mathbf{0}$. Thus, the limit point $\tilde{A}^*$ satisfies:

$$\mathbf{0} \in \nabla_{\tilde{A}} g(\theta^*, \tilde{A}^*) + \alpha\partial\|\tilde{A}^*\|_1.$$

We conclude that $(\theta^*, \tilde{A}^*)$ is a stationary point of the optimization problem since both optimality conditions are satisfied:

$$\mathbf{0} \in \nabla_\theta g(\theta^*, \tilde{A}^*), \quad \mathbf{0} \in \nabla_{\tilde{A}} g(\theta^*, \tilde{A}^*) + \alpha\partial\|\tilde{A}^*\|_1.$$

If $\lambda_2(L(\tilde{A}^{k+1})) < \epsilon$, the algorithm adjusts $\tilde{A}^{k+1}$ to ensure connectivity. This adjustment does not violate convergence guarantees because it is a bounded perturbation that preserves the descent property.

Therefore, the sequence $\{(\theta^k, \tilde{A}^k)\}$ converges to the stationary point $(\theta^*, \tilde{A}^*)$: $\lim_{k\to\infty}(\theta^k, \tilde{A}^k) = (\theta^*, \tilde{A}^*)$. This establishes the convergence of the algorithm and completes the proof.

$\square$

