# OpenReview forum: "DeepNT: Path-Centric Graph Neural Networks for Network Tomography"
_ICLR.cc/2025/Conference — Submitted to ICLR 2025_

### Official Review · Reviewer_iHJU · 2024-10-21

**Soundness:** 2
**Presentation:** 3
**Contribution:** 2
**Rating:** 5
**Confidence:** 3

**Summary:**

This paper proposed Deep Network Tomography (DeepNT), a new framework that learns a path-centric graph neural network for predicting path performance metrics. The key contributions of this paper include (1) the path-centric graph neural network that infers the path performance metrics (PPM) values of a path by learning its path embedding, and (2) a learning objective that imposes connectivity and sparsity constraints on topology and path performance triangle inequality over PPMs.

**Strengths:**

1. The ideas in this paper are clearly expressed.
2. The experimental results demonstrate the improvements of the proposed method over existing approaches.
3. The paper presents interesting findings and contributes to the field of network tomography.

**Weaknesses:**

- DeepNT proposed in this paper suppose the adjacent matrix is unknown or incomplete. Then how to construct the path sets $P_{uv}$ that represent the set of all possible paths from the node $u$ to $v$ without a specific adjacent matrix? How do we know which path has the optimal path performance? If the path performance can be tested by sending numbers of end-to-end probes, should the probing overhead be considered before the model training? Additionally, how to control the probes to traverse the specific routing paths is also a tough question since many networks do not support source routing.
- On contribution of this paper is the design of the path performance triangle inequality. However, in many scenarios, the triangle inequality may not hold. For example, the delay between $<u,v>$ may not always be less than the delay of $<u,z> + <z,v>$ due to the load balancing policy in the current network. It is even less applicable in social networks.
- In the experiments, the comparison with the baselines may not be entirely fair. In the scenarios of social networks, it's more like doing a matrix completion task. The seven baselines in the experiments are all within the area of network performance tomography, which were originally designed for network performance measurement. Hence, a more reasonable way is to compare DeepNT with matrix completion methods in such a scenario.
- Network topology reconstruction is an important contribution in this paper. However, the experiments did not fully demonstrate the advantages in this part. The author only carried out the experiments when the topological error rate was set to 0.2. More experiments (e.g., higher missing rates or block missing) are required to validate the effectiveness of DeepNT.

**Questions:**

The questions I would like the authors to address have already been raised in the "Weaknesses" section. I hope the authors can provide more information on these aspects.

---

> ### Author Response · Authors · 2024-11-22
> **Response to reviewer iHJU**
>
> We really appreciate your thorough and constructive feedback. We are eager to respond to your questions and comments:
>
> ---
>
> **Response to W1**:
>
> > How to collect the path sets $P_{uv}$ when the adjacent matrix is unknown or incomplete? How do we know which path has the optimal path performance?
>
> - In line 145, $P_{uv}$ represents the set of all possible paths from $u$ to $v$, so it is constructed given the known topology. It is used to define the optimal path performance from $u$ to $v$, e.g., $y_{uv} = \{y_{uv}^n \mid y_{uv}^n \leq y_{uv}^k, ~\forall p_{uv}^k \neq p_{uv}^n \in P_{uv}\}$ for additive metrics. $P_{uv}$ does not need to be constructed in DeepNT.
>
> - $y_{uv}$ represents the observed optimal path performance from $u$ to $v$. The specific optimal path information between $<u,v>$ is unknown due to the unknown or incomplete topology. DeepNT aims to predict the optimal path performance. Thus, to predict $\hat y_{uv}$, DeepNT collects $P_{uv}^L$ (line 200) using the observed network topology ($A$) and path performance collection ($S$) then aggregate their information.
>
> > How about the probing overhead? How to control the probes to traverse the specific routing paths is also a tough question since many networks do not support source routing?
>
> Since DeepNT relies solely on observations of path performance, such as delay in the Internet, to serve as the training data, no probing is required during or prior to training.
>
> ---
>
> **Response to W2**: As mentioned in W1, DeepNT aims to predict the "optimal path performance" between nodes. For additive metrics such as delay, "optimal" implies that smaller values are better. If the ground truth optimal path performance $y_{uz}$ and $y_{zv}$ is observed for $<u,z>, <z,v> \in S$, then $\hat y_{uv} \leq y_{uz} + y_{zv}$. If this triangle inequality is violated, the predicted path performance for $<u,v>$ is no longer "optimal," as it would be worse than the performance of the path $u \rightarrow z \rightarrow v$. The triangle inequality ensures that if the complete optimal path for $<u,v>$ is not observed, the predicted path performance between them $\hat y_{uv}$ will not be worse than the performance of any other observed path connecting $u$ and $v$.
>
>
> - For static load balancing policies, the triangle inequality works for most of path performan metrics.
>
> - For dynamic load balancing policies or social networks, the triangle inequality holds by continuously updating the observation set $S$ and retraining DeepNT when necessary.
>
> ---
>
> **Response to W3**: We agree that low-rank approximation techniques, such as matrix completion and matrix factorization, can be applied to network tomography in certain scenarios. Matrix factorization, in particular, has been explored in network tomography due to the inherent low-rank properties of network performance metrics. While matrix factorization explicitly models this low-rank structure, making it well-suited for reconstructing missing or incomplete data—particularly in the context of sparse matrices—it is no longer a commonly used technique in network tomography. This is primarily because low-rank approximation techniques require a substantial amount of observed data to achieve robust performance.
>
> Empirically, we compared DeepNT against two widely used matrix factorization techniques—Non-negative Matrix Factorization (NMF) [1] and Neural Matrix Factorization (NeuMF) [2]—for predicting additive metrics in terms of MAPE (%):
>
> | $\frac{\|S\|}{\|T\|}$| NMF | NeuMF | DeepNT|
> | --- | --- | --- | :---: |
> | 10% | 89.06 | 76.13 | 25.20  |
> | 20% | 68.83 | 60.77 | 21.68 |
> | 30% | 57.05 | 54.62 | 19.35  |
>
> - DeepNT consistently and significantly outperforms matrix factorization techniques. Particularly, when $\frac{\|S\|}{\|T\|}$ = 10% (i.e., only 10% of node pairs' path performance is observed), the predictions by both NMF and NeuMF become unreliable.
>
> - The performance of matrix factorization methods drop significantly as fewer path performances are available. The low-rank modeling relies heavily on observing a substantial amount of data (e.g., path performance between many node pairs) to ensure reliable predictions.
>
> ---
>
> **Response to W4**: We conducted experiments with higher missing rates in the adjacency matrix. The main results are available at: https://anonymous.4open.science/r/DeepNT2-1D25/topology.png. DeepNT demonstrates promising performance even when the topology missing rate approaches 0.7.
>
> ---
>
> Thank you for your valuable effort in evaluating our work! We are happy to provide further clarification and address any additional concerns that may influence your assessment of its suitability for acceptance.
>
> ---
>
> **References**
>
> [1] Lee, Daniel, and H. Sebastian Seung. "Algorithms for non-negative matrix factorization." Advances in neural information processing systems 13 (2000).
>
> [2] He, Xiangnan, et al. "Neural collaborative filtering." Proceedings of the 26th international conference on world wide web. 2017.

---

> > ### Comment · Reviewer_iHJU · 2024-11-24
> >
> > Thank the author for the response; the experimental results looks great. However, I still have concerns about the current presented version:
> > - The effectiveness of the method described in this paper relies heavily on the training data, which may not necessarily align with the triangular optimality assumption mentioned in the main text.
> > - Continuously updating data implies the need to recollect training data, yet, as mentioned in **Response to W1**, this paper does not provide a relevant solution. Moreover, in the context of social networks, how exactly does the triangular inequality come into play?
> > - Which dataset does **Response to W3** use? Are the experimental settings fair? To my knowledge, many methods can handle over 90% missing data.

---

> > > ### Author Response · Authors · 2024-12-01
> > > **Could you let us know if our rebuttal has sufficiently addressed your concerns?**
> > >
> > > Dear Reviewer iHJU,
> > >
> > > We recognize that the timing of this discussion period may not align perfectly with your schedule, yet we would greatly value the opportunity to continue our dialogue before the deadline approaches.
> > >
> > > We hope that our responses and additional experiments have effectively addressed your concerns. We truly appreciate all the valuable advice we have received. Could you let us know if your concerns have been adequately addressed? If you find that your concerns have been resolved, we would appreciate it if you could reconsider the review score.
> > >
> > > Thanks!

---

> > > > ### Author Response · Authors · 2024-12-02
> > > > **A friendly reminder for further discussion**
> > > >
> > > > Dear Reviewer iHJU,
> > > >
> > > > We hope this message finds you well. The rebuttal phase ends today and we would like to know if our further response has completely addressed your concerns. In our further response, we **have provided explanations and examples of the Triangle Inequality**. Furthermore, **we would be glad to include a comparison with matrix factorization methods that you believe can achieve competitive performance under conditions of sparse observations**. We would really appreciate that if you could check our response. If you find that your concerns have been resolved, we would appreciate it if you could reconsider the review score. Looking forward to hearing back from you.
> > > >
> > > > Best Regards,
> > > >
> > > > Authors

---

> ### Author Response · Authors · 2024-11-24
> **Follow-up response to reviewer iHJU**
>
> Thank you for giving us the opportunity to provide further clarifications!
>
> ---
>
> > The effectiveness of the method described in this paper relies heavily on the training data, which may not necessarily align with the triangular optimality assumption mentioned in the main text.
>
> **Response**: Actually, as long as one end-to-end path performance metric has a definition of "optimality," our method can be applied effectively, regardless of the type of the metrics. There may be some misunderstanding about the task we are addressing. To clarify, a specific example is provided in the response to your second question below.
>
> ---
>
> > Continuously updating data implies the need to recollect training data, yet, as mentioned in **Response to W1**, this paper does not provide a relevant solution. Moreover, in the context of social networks, how exactly does the triangular inequality come into play?
>
> **Response**:
>
> - **On Continuously Updating Data and Recollecting Training Data**: To clarify, the main contribution of our work is to enhance network tomography performance, reduce training costs, and adapt to any path performance metrics. It is important to highlight that the challenge of data collection is not unique to our method; it is a common issue faced by all network tomography approaches and does not diminish the significance of our contribution to the community.
>
> - **On the Role of Triangular Inequality in Social Networks**: When there is a need to predict optimal end-to-end performance between users in social networks, DeepNT, leveraging the triangle inequality, is well-suited for this purpose. Below, we provide a simple example to illustrate its applicability (assume there are 3 users $A$, $B$, and $C$):
>   - Suppose $A$ and $C$ have an optimal trust score of $T(A,C) = 0.8$, but the link between $<A,C>$ is not observed (i.e., incomplete topology). However, there are observed links between $<A, B>$ and $<B, C>$. The trust score between the two unlinked nodes is inferred through the nodes linking them, using a multiplicative form: $T(A,C) = T(A,B) \times T(B,C) = 0.7 \times 0.5 = 0.35$.
>   - When predicting the optimal trust score for $<A, C>$, if the path $A \rightarrow B \rightarrow C$ is the observed optimal path, the predicted trust score will be constrained by the triangle inequality, ensuring it is no worse than $0.35$.
>   - It might be assumed that $T(A,C) < 0.35$ such as 0.2, which would appear to violate the constraint. However, since we are predicting the **"optimal"** end-to-end performance, the triangle inequality still holds, as the optimal performance between $<A,C>$ now is 0.35 but not 0.2.
>
> ---
>
> > Which dataset does **Response to W3** use? Are the experimental settings fair? To my knowledge, many methods can handle over 90% missing data.
>
> **Response**:
> - **Dataset**: The experiments were conducted on the synthetic dataset, including Erdős-Rényi, small-world, and power-law graphs.
> - **Performance**: The comparison was performed under the condition that 10% of node pairs had observed end-to-end performances, ensuring a fair comparison.
>   - Both prior studies, such as [1], and our experimental results demonstrate that low-rank approximation techniques, such as Matrix Factorization, fail to ensure satisfactory performance in network tomography when the observed data is sparse.
>   - For instance, as highlighted in [1], the correlation of link metric predictions decreases as fewer end-to-end path performance is utilized, indicating reduced estimation accuracy due to the limited availability of data. It demonstrates good performance only when more than 70% of end-to-end performance data is observed.
>   - Could you provide specific references where Matrix Factorization has been reported strong performance under conditions of sparse observations? We would be happy to reproduce their results and compare them with our DeepNT as soon as possible.
>
> ---
>
> Thank you for your valuable feedback! We are happy to provide further clarification and address any additional concerns that may influence your assessment of its suitability for acceptance.
>
> ---
>
> **References**:
>
> [1] Raza, Muhammad H., et al. "Network tomography by non negative matrix factorization (NNMF)." Proceedings of the 2010 International Symposium on Performance Evaluation of Computer and Telecommunication Systems (SPECTS'10). IEEE, 2010.

---

### Official Review · Reviewer_XbAT · 2024-10-28

**Soundness:** 3
**Presentation:** 3
**Contribution:** 3
**Rating:** 6
**Confidence:** 3

**Summary:**

This article introduces a new framework named DeepNT, aimed at solving the challenges in network tomography by integrating path-centric GNNs. Network tomography involves using observable PPMs to infer network characteristics. The authors point out that traditional methods, which rely on a complete understanding of network topology, and recent purely data-driven approaches both have their limitations. In contrast, DeepNT combines data-driven techniques with appropriate inductive biases from partial prior knowledge.

**Strengths:**

1. The DeepNT model introduces a path-centric GNN, providing a new theoretical framework for network tomography. Unlike traditional GNNs that focus on nodes or entire graphs, the path-centric GNN concentrates on learning the dynamic performance of networks through the aggregation of information from sequences of nodes. This method allows the model to capture interactions and dependencies along paths more intricately, thereby enabling more accurate predictions of PPMs (Path Performance Metrics) that are not directly observable in the network.
2. During its training process, the DeepNT model integrates structural constraints with performance constraints. The core idea of this design philosophy is to merge the fundamental principles of network theory with the optimization process of machine learning, aimed at ensuring that the learned network topology not only complies with physical or logical feasibility but also reflects the actual network performance. This hybrid approach enhances the model's ability to infer the intrinsic structure and performance of the network.
3. DeepNT exhibits a clear advantage in predicting path performance metrics compared to other existing state-of-the-art technologies. Moreover, DeepNT demonstrates particular robustness when dealing with large-scale and complex network structures. The experimental design of DeepNT takes into account various factors such as latency, bandwidth, packet loss rate, different network topologies, and various network operation scenarios. Furthermore, the experiments utilize multiple evaluation metrics to ensure the accuracy and reliability of the results

**Weaknesses:**

1. Pooling operations, although helpful in simplifying data and reducing dimensions, can also lead to the loss of important information. Particularly in cases where paths are long or there is significant variation in node features, a single pooling operation may not be sufficient to capture all critical feature information, limiting the model's flexibility and accuracy in handling different types of network structures.
2. Network states and structures in real-world applications are often dynamic, including the dynamic addition or removal of nodes and edges, as well as real-time changes in network traffic and congestion. These dynamic characteristics require network tomography models to be highly flexible and rapidly adaptable. If the DeepNT model relies primarily on static graph structures and previously learned features, its ability to quickly adapt to such dynamic changes may be limited. Especially when the model depends on long training periods and extensive historical data to predict network performance accurately, its responsiveness and accuracy in dealing with real-time or near-real-time network state changes may be inadequate.
3. The DeepNT model does not consider the sequentiality of node paths, i.e., the order of nodes in the path, which might affect the model's performance in network tomography. Ignoring the sequentiality could lead to inaccurate predictions of certain network performance metrics.
4. The DeepNT model trains using only a portion of the information available in the graph, which may result in poor performance when the model encounters network configurations or conditions that are not represented in the training data, i.e., poor generalization.

**Questions:**

See the above weakness.

---

> ### Author Response · Authors · 2024-11-22
> **Response to reviewer XbAT**
>
> We sincerely thank the Reviewer XbAT for acknowledging our contributions to network tomography and providing insightful observations.
>
> ---
>
> **Response to W1**: To clarify, DeepNT employs a path-centric aggregation layer that considers multiple promising paths rather than relying on a single pooling operation. This design effectively captures diverse contextual information and mitigates the loss of important details. Towards your specific concerns:
> > **What if paths are long?** As described in line 200, DeepNT samples paths with lengths not exceeding $L$. This ensures that the aggregation process retains as many local features as possible while avoiding the over-smoothing problem encountered in long path aggregation.
>
> > **What if there is significant variation in node features?** In network tomography, node features are typically derived from the topological structure, using methods such as one-hot encoding or deep walks. These approaches standardize the embedding spaces across different networks, minimizing the issue of varying node feature distributions. DeepNT has demonstrated state-of-the-art performance across networks with diverse structures, such as Internet and social networks.
>
> ---
>
> **Response to W2**: To clarify, the primary goal of network tomography is to infer and predict network dynamics—such as traffic patterns and congestion—using end-to-end measurements. This inherently involves collecting and analyzing a significant amount of data. Below, we address your specific concerns:
>
> > How to deal with the dynamic addition or removal of nodes and edges?
> - **Nodes**: The network tomography problem typically assumes that the set of nodes is fixed and known beforehand. This assumption is a standard limitation across traditional and modern approaches.
> - **Edges**: For edge dynamics, DeepNT is capable of learning the network topology dynamically, given the set of nodes. This enables the model to infer missing edges effectively without impacting its performance.
> - **Our Improvement**: Traditional network tomography methods assume both the set of nodes and the network topology are fully known. DeepNT relaxes this requirement by only needing knowledge of all nodes, which represents a significant step forward.
>
> > Inadequate ability to handle real-time or near-real-time network dynamics:
> - **Real-time scenarios**: Gathering sufficient end-to-end measurements in real-time is inherently challenging. It often requires probing across multiple paths or waiting for traffic patterns to stabilize, which limits the applicability of network tomography to fully real-time scenarios.
> - **Near-real-time scenarios**: DeepNT demonstrates strong potential for near-real-time analysis by significantly reducing the training complexity from $O(|V|^3)$, as seen in traditional network tomography methods, to $O(|E|)$. This not only enhances DeepNT's scalability but also enables it to quickly update training data and retrain the model, making near-real-time class prediction feasible—a task that is impractical for traditional methods due to their high computational cost.
>
> ---
>
> **Response to W3**: DeepNT does not prioritize sequentiality, as its primary focus is on capturing and utilizing potential nodes that contribute to forming the optimal path between two nodes from multiple promising paths, rather than strictly adhering to the order of nodes within each path. This design aligns with the core objective of network tomography: to infer performance metrics by leveraging patterns in the graph structure rather than the specific traversal order of nodes in individual paths.
>
> ---
>
> **Response to W4**: We agree that generalization is a critical aspect of network tomography. To clarify, DeepNT is specifically designed to address the challenges of learning from incomplete graph information, as partial observability is a defining characteristic of real-world network tomography scenarios.
>
> To ensure the generalization of DeepNT, we have incorporated the following key designs:
> - Path-Centric Aggregation: DeepNT aggregates information from multiple promising paths. By sampling multiple loopless paths of limited length, DeepNT incorporates both local and global structural information, enhancing its generalization .
> - Learning Topology Dynamically: DeepNT dynamically infers the missing topology during training, allowing the model to effectively reconstruct the network structure and improve generalization.
> - Triangle Inequality Constraint: The triangle inequality constraint introduces a critical inductive bias that helps the model make logical and consistent predictions. By enforcing that path performance predictions respect inherent relationships between nodes and paths, this constraint ensures more reliable generalization.
>
> ---
>
> Thank you for your valuable effort in evaluating our work!  We are happy to provide further clarification and address any additional concerns that may influence your assessment of its suitability for a clear acceptance.

---

> > ### Comment · Reviewer_XbAT · 2024-11-25
> >
> > Thank you for your response. I greatly appreciate and commend the effort the authors have invested during the rebuttal phase. However, after carefully considering the current completeness of the manuscript, the feedback from other reviewers, and the rebuttal content, I am inclined to maintain the current score. I encourage the authors to further refine their work and wish them success in future submissions.

---

> ### Author Response · Authors · 2024-12-01
> **Thanks for the Reviewer’s Constructive Feedback**
>
> Thank you for your thoughtful and constructive feedback. We are committed to incorporating the suggested changes by all reviewers in our revisions to further enhance the manuscript.
>
> As suggested by other reviewers, We have extended our ablation study to evaluate alternative aggregation layers and graph structure learning methods and provided detailed analysis of the expressive power of our model. We would appreciate it if you could let us know before closing the discussion whether there are still concerns that may influence your assessment of its suitability for a clear acceptance!

---

### Official Review · Reviewer_MzbY · 2024-10-31

**Soundness:** 2
**Presentation:** 2
**Contribution:** 3
**Rating:** 5
**Confidence:** 3

**Summary:**

The paper presents DeepNT, a path-centric graph neural network (GNN) framework for network tomography that predicts unobserved path performance metrics (PPMs), such as latency and bandwidth, by leveraging partially observed network data. DeepNT introduces a unique approach by combining data-driven learning with partial prior knowledge, addressing limitations in traditional methods that require complete topology knowledge. Key contributions include a path-centric GNN that aggregates node embeddings along paths, as well as connectivity, sparsity, and triangle inequality constraints that enhance the accuracy of inferred topologies and metrics. Extensive experiments demonstrate that DeepNT outperforms existing methods in predicting multiple types of PPMs across real-world and synthetic datasets, showing strong scalability and robustness.

**Strengths:**

1. The proposed DeepNT can effectively integrate connectivity, sparsity and triangle inequality constraint to enable robust inference on network topology and path performance metrics under the incomplete data setting.

**Weaknesses:**

1. The paper currently limits node embedding to path-centric methods and relies primarily on DeepWalk-inspired random walks and SVD-based HOPE embeddings. This focus might overlook more complex embeddings, like graph attention or spectral approaches, which could capture different node relationships and further improve performance.
2. While DeepNT demonstrates favorable scalability in empirical evaluations, the paper could benefit from a deeper theoretical analysis of its computational complexity, particularly regarding training on large-scale graphs. A formal complexity analysis for different network sizes, embedding types, and constraint applications would clarify computational trade-offs, especially since training with large adjacency matrices and triangle inequality constraints may become a bottleneck.

**Questions:**

1. While the article's method of updating the adjacency matrix by imposing constraints on connectivity and sparsity appears simple and effective, it lacks necessary theoretical discussions, such as an analysis of overall training complexity, convergence studies, and comparisons of computational efficiency with previous methods.
2. In the synthetic graph experiments, it would be beneficial to test other topologies beyond Erdős-Rényi graphs, such as small-world and power-law graphs.

---

> ### Author Response · Authors · 2024-11-22
> **Response to reviewer MzbY**
>
> We really appreciate your thorough and constructive feedback. We have taken your suggestions to heart and made changes to the best of our abilities.
>
> ---
>
> **Response to W1**: To clarify, we tested various encoding methods for node embedding initialization, such as one-hot encoding, Node2Vec, and random walks since graphs are often featureless in typical network tomography scenarios. They are only used for node embedding initialization. Our proposed framework, DeepNT, already integrates a path-centric graph neural network to further learn node embeddings dynamically during training. This allows us to leverage both the initialization and the flexibility of GNN architectures. Additionally, the framework is adaptable to different types of GNN backbones, including attention-based (e.g., GAT, GraphTransformer) and spectral-based (e.g., GCN) approaches.
>
> We acknowledge that showing the performance of using different GNN models can help demonstrate the robustness of DeepNT to GNNs. The MSE results for additive metrics when $\frac{|S|}{|T|}$=30% using different GNN as the backbone, including GAT, GraphTransformer, and GCN are summarized in the following Table.
>
> | Dataset | GAT | GCN | Graph Transformer |
> | --- | --- | --- | :---: |
> | Internet | 71.19 | 71.08 | 70.10 |
> | Social Network  | 10.26 | 10.81 | 10.55 |
> | Transportation  | 17.94 | 18.66 | 18.25 |
> | Synthetic  | 61.73 | 59.04 | 60.28 |
> ---
>
> **Response to W2**: Thank you for your valuable comment! Below, we provide a detailed complexity analysis of DeepNT, emphasizing its scalability in sparse large-scale graph scenarios, which are typical in network tomography applications.
>
> Due to the sparsity of real-world networks, we have $|E| << |V|^2$, which makes sparse matrix multiplications during GNN operations have a complexity of $O(|E| \cdot d)$, not $O(|V|^2 \cdot d)$, where $d$ is the embedding dimension. For the triangle inequality constraint, instead of processing all possible triplets $\(O(|V|^3\)$, we sample a subset of triplets proportional to $|E|$. This reduces the complexity to $O(t\cdot k\cdot |E|)$, where $k$ is the sampling ratio and $t$ is the number of training iterations.
>
> | **Component**              | **Complexity**                                       |
> |---|---|
> | Sparse Matrix Multiplication | $O(\|E\| \cdot d)$                                   |
> | Triangle Constraints        | $O(t \cdot k \cdot \|E\|)$                            |
> | L1 Regularization           | $O(\|E\|)$                                            |
> | Connectivity Constraints           | $O(\|E\| + \|V\|)$                                            |
>
> There are no $|V|^2$ or $|V|^3$ terms in the complexity, providing a theoretical guarantee of DeepNT’s scalability on large-scale sparse graphs.
>
> ---
>
> **Response to Q1**:
> > **Training Complexity**: Given that $|E| << |V|^2$ and the complexity of key components above, the GNN training complexity and the  adjacency matrix update complexity are both dominated by $O(|E|)$.
>
> > **Convergence Analysis**: The proof of convergence for our optimization algorithm is available at: https://anonymous.4open.science/r/DeepNT2-1D25/Convergence.pdf.
>
> > **Computational Efficiency**: Classical network tomography approaches rely on solving linear systems has the complexities of $O(|V|^3)$ or higher. Deep Learning-Based Methods such as NeuTomography avoid direct adjacency matrix updates but require dense path sampling and augmentation, leading to the complexity of $O(|V|^2)$. In contrast, DeepNT achieves a significantly lower complexity of $O(|E|)$.
>
> ---
>
> **Response to Q2**: We have now incorporated experiments on other types of synthetic graphs, including small-world and power-law graphs. Specifically, we use the Watts-Strogatz model to generate small-world graphs and the Barabási-Albert model to generate power-law graphs. We have gathered our results in Table 2, 3, 4, 5:
>
> - DeepNT consistently outperforms all other comparison methods.
>
> - The results indicate that DeepNT performs better on small-world graphs and Erdős-Rényi graphs compared to power-law graphs. This may be because, in power-law graphs, paths often pass through the same hubs, and missing observations from these critical hubs can disrupt predictions.
>
> ---
>
> Thank you for your valuable effort in evaluating our work! We are happy to provide further clarification and address any additional concerns that may influence your assessment of its suitability for acceptance.

---

> ### Author Response · Authors · 2024-12-01
> **Could you let us know if our rebuttal has sufficiently addressed your concerns?**
>
> Dear Reviewer MzbY,
>
> We recognize that the timing of this discussion period may not align perfectly with your schedule, yet we would greatly value the opportunity to continue our dialogue before the deadline approaches.
>
> We hope that our responses and additional experiments have effectively addressed your concerns. We truly appreciate all the valuable advice we have received. Could you let us know if your concerns have been adequately addressed? If you find that your concerns have been resolved, we would appreciate it if you could reconsider the review score.
>
> Thanks!

---

> > ### Author Response · Authors · 2024-12-02
> > **Looking forward to your reply**
> >
> > Dear Reviewer MzbY,
> >
> > We hope this message finds you well. The rebuttal phase ends today and we would like to know if our further response has completely addressed your concerns. In our response, we **have tested various GNNs to capture different node relationships** and **conducted a detailed theoretical analysis covering complexity, optimization convergence, and expressive power.** We believe that we have addressed your previous concerns. We would really appreciate that if you could check our response. If you find that your concerns have been resolved, we would appreciate it if you could reconsider the review score. Looking forward to hearing back from you.
> >
> > Best Regards,
> >
> > Authors

---

### Official Review · Reviewer_Nzqt · 2024-11-04

**Soundness:** 3
**Presentation:** 2
**Contribution:** 2
**Rating:** 5
**Confidence:** 4

**Summary:**

This paper presents a neural approach for network tomography called DeepNT. The proposed approach jointly learns the network topology and the GNN for predicting path metrics using alternating optimization. In particular, the path metrics are predicted based on a path aggregation layer that is trained to predict pairwise metrics based on observations. Moreover, triangle inequality is incorporated into the loss to enforce the best path to be selected (according to some objective metric). The graph is learned as an adjacency matrix with an L1 penalty and a connectivity constraint based on its second eigenvalue. The authors propose an alternating optimization scheme (GNN and graph learning) that uses projection to minimize L1 and guarantee a valid matrix. Ten datasets and six baselines are considered in the experiments. The results show that DeepNT consistently outperforms the baselines.

**Strengths:**

- The paper addresses a relevant problem
- Multiple classical approaches are considered as baselines
- The empirical results described are promising

**Weaknesses:**

- The contributions of the paper are limited
- Some important baselines are missing
- The evidences supporting the proposed method are purely empirical


Detailed comments:

1) Contributions: graph learning has already been extensively studied in the context of GNNs and the proposed path aggregation layer does not seem particularly novel.

2) The proposed graph learning approach could be combined with any GNN in the literature. It is not clear that the proposed path aggregation is significantly better than typical GNN aggregation or the subgraph aggregation using by link prediction methods such as SEAL. Moreover, there are many graph learning approaches, including bilevel optimization, that are not compared against the one proposed in the paper.

3) Supporting evidence for proposed solution: while the results presented are promising, it would be important to provide some theoretical support or at least better intuition on why NeuroNT is a good solution. For instance, are there any guarantees that the predicted pairwise metrics are near optimal given enough observations? This would definitely strengthen the paper.


** Comments after the rebuttal:
I'm increasing my score for this paper based on the most recent reviews. The updated results show that the baselines considered earlier were not good enough. The original paper didn't cite the path neural networks paper but now uses its expressiveness proof. The related work and contributions of the paper need to be adjusted accordingly and the path neural networks approach should be included as a baseline.

**Questions:**

- What are the major contributions of the paper?

- What other baselines could be considered from the GNN literature?

- Is there any theoretical evidence supporting the proposed solution?

**Details Of Ethics Concerns:**

I did not identify any ethics issue with this paper.

---

> ### Author Response · Authors · 2024-11-22
> **Response to reviewer Nzqt (1/2)**
>
> We would like to thank you for your thorough review and interesting thoughts. In the following, we attempt to answer your questions and comments:
>
> ---
>
> **Response to W1 & Q1**: While we acknowledge that graph learning and Graph Neural Networks (GNNs) are well-established, our work distinguishes itself by addressing critical challenges specific to the network tomography domain through the following novel contributions:
>
> - **Towards the graph learning**: We first highlight and summarize our contributions to graph learning as follows:
>   - Introducing a unified optimization framework: We propose a novel method that simultaneously learns both the metric prediction model and the graph structure, integrating additonal unlabeled supervision by the triangle inequality. In the subsequent responses to W2 and W3, we will prove that our optimization algorithm has good efficiency and theoretically guaranteed convergence.
>   - Adapting graph learning to more challenging scenarios: DeepNT is specifically designed for network tomography, where less structural information and training data are available compared to standard graph learning tasks such as node classification.
>
> - Most graph structure learning methods assume that the adjacency matrix is either complete with noise or partially observed. These approaches primarily focus on refining or denoising the graph structure rather than reconstructing it entirely. In contrast, DeepNT addresses the challenging setting of network tomography, where the graph topology may be entirely unknown, and only a small fraction of pairwise metrics is observed (e.g., when only 10% of end-to-end path performance metrics are available). This scenario necessitates a fundamentally different approach, including the joint learning of the adjacency matrix and path-centric metrics, guided by additional supervision through constraints, i.e., the proposed triangle inequality.
> - **Towards the path aggregation layer**: The proposed path aggregation layer is a key component tailored to the network tomography setting. Standard GNN aggregation methods (e.g., sum, mean, attention) operate on local neighborhoods and cannot capture dependencies between nodes connected by longer paths. In contrast, the path aggregation layer explicitly aggregates embeddings across multiple candidate paths, effectively capturing promising nodes that may constitute the optimal path between node pairs, specifically addressing the path-centric demands of network tomography.
>
> ---
>
> **Response to W2 & Q2**: Thanks a lot for you helpful comment!
>
> > How about other GNN aggregation methods such as the subgraph aggregation like SEAL?
>
> SEAL's framework is specifically designed for link prediction tasks [1], which focus on predicting whether a specific edge exists between two nodes. While SEAL is highly effective in capturing the local structure around links, its direct applicability to network tomography is limited. In network tomography, understanding path interactions across the network is critical, which SEAL does not inherently support. If the hop number is small, SEAL cannot capture sufficient path information, while increasing the hop number introduces more irrelevant nodes, adversely impacting the prediction of path performance. Empirically, we replaced our path aggregation with SEAL's subgraph aggregation for predicting additive metrics. The results of MSE for predicting addtive metrics when $\frac{|S|}{|T|}$ = 30% are shown as follows,
>
> | DeepNT | Internet | Social Network | Transportation | Synthetic |
> | ----------------- | -------- | -------------- | -------------- | --------- |
> | w/ path aggregation | 71.08  | 10.81  | 18.66   | 59.04  |
> | w/ subgraph aggregation |   117.65   | 22.90  |  21.77  |  77.93  |
>
> Although the original SEAL paper only sets the hop number $h$ as 1 or 2, we extended our evaluation to test $h$ up to 10 and selected the best-performing value. This adjustment accounts for the longer possible path lengths between nodes, ensuring a fair comparison.
>
> > How about other graph learning approaches such as bilevel optimization?
>
> We compared our unified optimization framework with the bilevel optimization approach [2]. In the bilevel framework, constraints of connectivity and sparsity are applied to the outer objective, while the GNN loss serves as the inner objective. The MAPE results (in %) for predicting additive metrics when $\frac{|S|}{|T|}$ = 30% are presented as follows:
>
> | DeepNT  | Transportation | Synthetic |
> | -- | -- | -- |
> | w/ unified optimization  | 37.94 | 19.35 |
> | w/ bilevel optimization |  44.91  | 22.97 |
>
> Our unified optimization approach outperforms the bilevel optimization framework in DeepNT. Notably, our unified optimization is significantly more efficient in achieving convergence, as it combines all objectives into a single unified framework.

---

> ### Author Response · Authors · 2024-11-22
> **Response to reviewer Nzqt (2/2)**
>
> **Response to W3 & Q3**: Thank you for your constructive comment. While the empirical results demonstrate the effectiveness of DeepNT, we are pleased to provide additional theoretical guarantees to further substantiate why DeepNT is a promising approach:
>
> - **Complexity**: Due to the sparsity of real-world networks, we have $|E| << |V|^2$, which makes sparse matrix multiplications during GNN operations have a complexity of $O(|E| \cdot d)$, not $O(|V|^2 \cdot d)$, where $d$ is the embedding dimension. For the triangle inequality constraint, instead of processing all possible triplets $\(O(|V|^3\)$, we sample a subset of triplets proportional to $|E|$. This reduces the complexity to $O(t\cdot k\cdot |E|)$, where $k$ is the sampling ratio and $t$ is the number of training iterations. As summarized below, the complexity does not include $|V|^2$ or $|V|^3$ terms, as seen in traditional approaches, providing a theoretical guarantee of DeepNT's scalability on large-scale sparse graphs.
>
> | **Component**              | **Complexity**                                       |
> |---|---|
> | Sparse Matrix Multiplication | $O(\|E\| \cdot d)$                                   |
> | Triangle Constraints        | $O(t \cdot k \cdot \|E\|)$                            |
> | L1 Regularization           | $O(\|E\|)$                                            |
> | Connectivity Constraints           | $O(\|E\| + \|V\|)$                                            |
>
> - **Proof of Convergence for the Unified Optimization Framework**: The proof of convergence for our optimization algorithm is available at: https://anonymous.4open.science/r/DeepNT2-1D25/Convergence.pdf.
>
> ---
>
> We have taken your suggestions to heart and made changes to the best of our abilities. We are happy to provide further clarification and address any additional concerns that may influence your assessment of its suitability for acceptance.
>
> ---
>
> **References**
>
> [1] Zhang, Muhan, and Yixin Chen. "Link prediction based on graph neural networks." Advances in neural information processing systems 31 (2018).
>
> [2] Franceschi, Luca, et al. "Learning discrete structures for graph neural networks." International conference on machine learning. PMLR, 2019.

---

> > ### Comment · Reviewer_Nzqt · 2024-11-25
> > **Response to rebuttal**
> >
> > Thanks for the responses to my comments.
> >
> > The authors have attempted to address each of my comments but I still believe a lot of work has to be done to support the novelty of the proposed approach. This is the case because ICLR is a venue focused on the model and its novelty, while a networking venue might be more interested in the experiments against traditional baselines. More specifically, an ablation study replacing the graph learning and the path aggregation schemes with existing solutions is needed. For the bilevel optimization (BO) results, it is not clear what "unified approach" means, BO is an alternative (there are others) to the graph learning approach (one can replace the other). Similarly, SEAL and other GNN aggregation methods are possible replacements for the path aggregation model. The new theorem provided is about the convergence of the model but does not show why NeuroNT is a better alternative than existing GNNs for the problem studied in this paper. In the graph learning community, we compare models using notions such as expressive power to say what a model can or cannot learn.
> >
> > I plan to keep my recommendation for this paper but I will participate in the discussion with other reviewers and the rebuttal will be considered during the discussion.

---

> > > ### Author Response · Authors · 2024-12-01
> > > **Could you let us know if our rebuttal has sufficiently addressed your concerns?**
> > >
> > > Dear Reviewer Nzqt,
> > >
> > > We recognize that the timing of this discussion period may not align perfectly with your schedule, yet we would greatly value the opportunity to continue our dialogue before the deadline approaches.
> > >
> > > We hope that our responses and additional experiments have effectively addressed your concerns. We truly appreciate all the valuable advice we have received. Could you let us know if your concerns have been adequately addressed? If you find that your concerns have been resolved, we would appreciate it if you could reconsider the review score.
> > >
> > > Thanks!

---

> > > > ### Author Response · Authors · 2024-12-02
> > > > **A friendly reminder for further discussion**
> > > >
> > > > Dear Reviewer Nzqt,
> > > >
> > > > We hope this message finds you well. The rebuttal phase ends today and we would like to know if our further response has completely addressed your concerns. In our further response, we **have expanded our ablation study** by comparing our model with currenly widely used aggregation layers and graph structure learning methods. Additionally, we **have conducted an analysis of the expressive power** of our model. We believe that we have addressed your previous concerns. We would really appreciate that if you could check our response. If you find that your concerns have been resolved, we would appreciate it if you could reconsider the review score. Looking forward to hearing back from you.
> > > >
> > > > Best Regards,
> > > >
> > > > Authors

---

> ### Author Response · Authors · 2024-12-01
> **Follow-up Response to reviewer Nzqt (1/2)**
>
> Thanks a lot for your valuable and constructive feedback. Your suggestions are very helpful in further improving the work, and we are refining the manuscript accordingly. We have extended our formal ablation study and added the analysis of the expressiveness of DeepNT model.
>
> ---
>
> > It is not clear what "unified approach" means.
>
> **Response:**
>
> The "unified approach" means the optimization framework proposed in our paper, integrating topology inference, path-metric prediction, and handling of partial data into a single, cohesive framework. This contrasts with disjointed methods that solve these problems independently like bilevel optimization.
>
> ---
>
> > More specifically, an ablation study replacing the graph learning and the path aggregation schemes with existing solutions is needed. For the bilevel optimization (BO) results, it is not clear what "unified approach" means, BO is an alternative (there are others) to the graph learning approach (one can replace the other). Similarly, SEAL and other GNN aggregation methods are possible replacements for the path aggregation model.
>
> **Response:** We have extended the ablation study by replacing the path aggregation layer with currently widely used aggregation layers and substituting our optimization algorithm with existing optimization techniques commonly employed in graph structure learning.
>
> - **Ablation Study on Aggregation Layer**: We compared the proposed path aggregation layer with aggregation operations in SEAL [1], LESSR [2], PNA [3], and OSAN [4].
>   - To provide a clearer explanation, we present the details of the aggregation layers along with their formulas and descriptions. The detailed descriptions are as follows:
> | **Model**               | **Formula**                                                                                                                                              | **Description**                                                                 |
> |----|---|---|
> | **DeepNT** | $$\hat h_v^{(n)} = h_v + \sigma(\sum_{z \in p_{uv}^{(n)}} \alpha_{vz}^{(n)} \cdot h_z^{(n)}))$$ $$\alpha_{vz}^{(n)} = \text{softmax}(r^\top [h_v, h_z^{(n)}])$$ $$h_v = \text{READOUT}(\hat h_v^{(n)}: p^{(n)}_{uv} \in \mathcal{P}^L_u)$$ | **Path aggregation**: Aggregates over sampled optimal paths using attention and combines node representations. |
> | **SEAL** | $$h_s = \text{AGG}(\{h_v : v \in G_h^{x, y}\})$$ $$h_{uv} = \text{READOUT}(\{h_s : G_h^{x, y} \in \text{enclosing subgraphs}\})$$ | **Subgraph aggregation**: Aggregates node features within enclosing subgraphs extracted for target links. |
> | **LESSR**| $$h_v = \text{GRU}(h_v, \{h_z : z \in \mathcal{N}(v)\})$$ $$h_{uv} = \text{Attention}(h_u, h_v, \text{shortcut connections})$$ $$h_v = \text{READOUT}(\{\text{EOPA}(h_v), \text{SGAT}(h_v)\})$$ | **Edge-order preserving aggregation**: Combines GRU-based local aggregation with global attention from shortcut paths. |
> | **PNA**                | $$h_{\text{agg}} = \text{AGG}(\{h_u : u \in \mathcal{N}(v)\})$$ $$h_{\text{scaled}} = \text{Scaler}(\deg(v)) \cdot h_{\text{agg}}$$ $$h_v = \text{Combine}(h_{\text{scaled}})$$ | **Principal neighbourhood aggregation**: Combines mean, max, min, and std aggregators with degree-based scaling.        |
> | **OSAN**               | $$h_s = \text{AGG}\left(\{f_{\text{WL}}(v) : v \in s\}\right)$$ $$h_v = \text{READOUT}\left(\{h_s : v \in s, s \in \mathcal{S}\}\right)$$         | **Ordered subgraph aggregation**: Aggregates features from WL-labeled subgraphs containing the node.             |
>   - We replaced our path aggregation layer with widely used aggregation methods to predict additive metrics. The Mean Squared Error (MSE) results for predicting additive metrics when $\frac{|S|}{|T|}$ = 30% are shown as follows,
> | DeepNT | Internet | Social Network | Transportation | Synthetic |
> | ---| ---| ---| ---| --- |
> | w/ path aggregation (ours) | **71.08**  | **10.81**  | 18.66   | **59.04**  |
> | w/ subgraph aggregation [1] |   117.65   | 22.90  |  21.77  |  77.93  |
> | w/ edge-order preserving aggregation [2] |  84.25  |  14.19  |   **17.11**  |  62.65  |
> | w/ principal neighbourhood aggregation [3] |  94.43  |  20.01  |  22.89   |   84.07 |
> | w/ ordered subgraph aggregation [4] |  101.73  |   17.26 |   20.57  |  71.14   |
>   - DeepNT with our path aggregation layer outperforms other DeepNT variants utilizing alternative aggregation layers across most datasets. This is because the path aggregation leverages path-specific attention on the most probable paths with optimal performance, enabling finer-grained and more accurate network tomography.

---

> ### Author Response · Authors · 2024-12-01
> **Follow-up Response to reviewer Nzqt (2/2)**
>
> - **Ablation Study on Structure Learning Algorithm**: We compared our structure learning method with other currently widely used structure learning algorithms, including Continuous Optimization [5], Bilevel Optimization [6], and Iterative Optimization [7, 8].
>   - The MAPE results (in %) for predicting additive metrics with $\frac{|S|}{|T|}$ = 30\% and a topology error rate of $\delta$ = 20\% are presented below:
> | DeepNT  | Transportation | Synthetic |
> | -- | -- | -- |
> | w/ unified optimization (ours)  | 37.94 | 19.35 |
> | w/ continuous optimization [5] |  46.26  |  23.88  |
> | w/ bilevel optimization [6] |  44.91  | 22.97 |
> | w/ iterative optimization [8] |  41.17  |  21.53  |
>   - DeepNT with our proposed structure learning algorithm outperforms its variants employing other structure learning methods, demonstrating the effectiveness of the triangle inequality constraint and the unified optimization framework.
>
> ---
>
> > In the graph learning community, we compare models using notions such as expressive power to say what a model can or cannot learn.
>
> **Response:**
>
> Thank you for your clarification! We have rigorously proved the expressive power of the proposed DeepNT model. The detailed proof is available at: https://anonymous.4open.science/r/DeepNT2-1D25/Expressiveness%20Study.pdf. Below is a summary of the key results:
> - DeepNT Expressiveness Beyond 1-WL.
>   - **Proof sketch**: We begin by introducing the concept of WL-trees, which are constructed by the 1-WL algorithm, and Path-trees, which are rooted at a node and constructed based on paths. We show that even when WL-trees are identical for two nodes, their corresponding Path-trees can still differ. Since DeepNT with the proposed path aggregation operates on Path-trees, it can distinguish nodes that 1-WL cannot, thereby completing the proof.
> - DeepNT Distinguishes Node Pairs Beyond 1-WL.
>   - **Proof sketch**: Consider two node pairs $(u, v)$ and $(u', v')$ with identical local neighborhoods up to 1-WL. If their path sets differ, i.e., $ P^L_{uv} \neq P^L_{u'v'}$, DeepNT assigns distinct embeddings: $f_{\text{DeepNT}}(u, v) \neq f_{\text{DeepNT}}(u', v')$. This distinctiveness is guaranteed by DeepNT’s path-centric aggregation, which encodes differences in path structures. Consequently, DeepNT surpasses the expressiveness of 1-WL in distinguishing node pairs.
> - Convergence of DeepNT Predictions to True Pairwise Metrics.
>   - **Proof sketch**: Leveraging the theorems on the expressiveness of DeepNT, all node pairs are uniquely identified and distinguished. Furthermore, the convergence of the optimization algorithm ensures that the training error diminishes and the generalization error diminishes as the sampled paths approach completeness. Consequently, DeepNT’s predictions converge to the true metrics in expectation with sufficient data and path sampling.
>
> ---
>
> If there are other papers you believe could provide alternative solutions, we would be happy to include a comparison with them. We are more than happy to address any additional concerns or questions that might impact your evaluation of the work's suitability for acceptance.
>
> ---
>
> **References:**
>
> [1] Zhang, Muhan, and Yixin Chen. "Link prediction based on graph neural networks." Advances in neural information processing systems 31 (2018).
>
> [2] Chen, Tianwen, and Raymond Chi-Wing Wong. "Handling information loss of graph neural networks for session-based recommendation." Proceedings of the 26th ACM SIGKDD international conference on knowledge discovery & data mining. 2020.
>
> [3] Corso, Gabriele, et al. "Principal neighbourhood aggregation for graph nets." Advances in Neural Information Processing Systems 33 (2020): 13260-13271.
>
> [4] Qian, Chendi, et al. "Ordered subgraph aggregation networks." Advances in Neural Information Processing Systems 35 (2022): 21030-21045.
>
> [5] Zheng, Xun, et al. "Dags with no tears: Continuous optimization for structure learning." Advances in neural information processing systems 31 (2018).
>
> [6] Franceschi, Luca, et al. "Learning discrete structures for graph neural networks." International conference on machine learning. PMLR, 2019.
>
> [7] Chen, Yu, Lingfei Wu, and Mohammed Zaki. "Iterative deep graph learning for graph neural networks: Better and robust node embeddings." Advances in neural information processing systems 33 (2020): 19314-19326.
>
> [8] Jin, Wei, et al. "Graph structure learning for robust graph neural networks." Proceedings of the 26th ACM SIGKDD international conference on knowledge discovery & data mining. 2020.

---

### Author Response · Authors · 2024-11-22
**Revision**

Dear reviewers,

we sincerely thank you for your thorough and detailed reviews. We appreciate the reviewers' recognition of the strengths of our work: its clear and well-written presentation (by Reviewers XbAT and iHJU), its promise for advancing network tomography (by Reviewers Nzqt and iHJU), and its strong scalability, robustness, and reliability (by Reviewers MzbY and XbAT).

We believe your suggestions have improved our paper. In the following, we present the changes made to the revision which are marked in Blue. We have also responded to each of your concerns under your respective reviews.

- We conducted more comprehensive experiments on synthetic data, considering a wider variety of graph types ( suggested by Reviewer MzbY).

- We have rewritten Section 4.4 to better emphasize our contributions to graph learning. A new Theorem 4.1, addressing the convergence of our optimization, has been introduced. The proof of Theorem 4.1 has been included in Appendix A.3 (suggested  by Reviewer Nzqt, MzbY).

- We have provided more comprehensive results for varying topology missing rates, as suggested by Reviewer iHJU, once the additional experiments are completed.

- We have extended our ablation study to evaluate alternative aggregation layers and graph structure learning methods, as detailed in Appendix A.4 (suggested by Reviewer Nzqt).

- Several new theorems have been introduced in our expressiveness study, with all proofs provided in Appendix A.5 (suggested by Reviewer Nzqt).

---

### Comment · Area_Chair_xBFe · 2024-11-23

Dear Reviewers,

The authors have uploaded their rebuttal and a revised version of the submission. Please take this opportunity to discuss any concerns you may have with the authors.

Best regards,
AC

---

### Meta-Review · Area_Chair_xBFe · 2024-12-20

**Metareview:**

The paper introduces DeepNT, a neural framework for network tomography that uses a path-centric graph neural network (GNN) to predict unobserved path performance metrics (PPMs) such as latency and bandwidth, based on partially observed network data. DeepNT combines data-driven learning with partial prior knowledge, addressing the limitations of traditional methods that require full topology knowledge. It employs a unique alternating optimization approach, incorporating a path aggregation layer and triangle inequality constraints to enhance the accuracy of predicted metrics. Extensive experiments demonstrate that DeepNT outperforms existing methods in predicting PPMs, showing scalability and robustness across real and synthetic datasets.

While the proposed DeepNT model shows promising results, the reviewers have identified several weaknesses that need to be addressed:

1. The paper lacks clarity regarding its problem positioning, making it difficult to determine its contribution either in the network tomography domain or in data imputation. This ambiguity weakens the paper’s overall impact, as it is not clear whether the work advances the field of network tomography or simply addresses a niche within data imputation.
2. The authors assume that data is already collected, which deviates from the original goal of network tomography methods, which aim to optimize data collection for better prediction accuracy. This assumption makes the work more similar to a data imputation task, rather than addressing the challenges of data collection and inference that are central to network tomography.
3. If positioned as a data imputation method, the proposed approach is outperformed by existing methods that can handle up to 90% missing data. This diminishes the contribution of the proposed method in the context of imputation tasks, as it does not provide a significant advantage over state-of-the-art techniques.

Based on these weaknesses, we recommend rejecting this paper. We hope this feedback helps the authors improve their paper.

**Additional Comments On Reviewer Discussion:**

In their rebuttal, the authors made several improvements, including conducting more comprehensive experiments on synthetic data with diverse graph types and extending their ablation study to evaluate alternative methods. They also clarified their contributions to graph learning, introduced a new convergence theorem, and provided additional results and theoretical studies as suggested by the reviewers.

However, during the rebuttal and discussion phase, the reviewers’ concerns regarding the assumption of pre-collected data, performance compared to existing methods, and lack of clarity in problem positioning remain unresolved. As a result, I recommend rejection based on the reviewers’ feedback.

---

### Decision · Program_Chairs · 2025-01-22

Reject